# Designing Observation and Action Models for Efficient Reinforcement Learning with LLMs

**Deunsol Yoon** [1]  **Sunghoon Hong** [1]  **Whiyoung Jung** [1]  **Junseok Park** [1]  **Geon-Hyeong Kim** [1]  **Woohyung Lim** [1]
**Soonyoung Lee** [1]  **Seung Hwan Kim** [1]  **Kanghoon Lee** [1]

## Abstract

Large Language Models (LLMs) have emerged as powerful tools for semantic reasoning, enabling the formalization of tasks that traditionally relied on manual human intuition. This capability extends to environment design in reinforcement learning. While prior research predominantly focuses on reward design, the design of observation and action spaces remains relatively underexplored. We propose LLM-based design of Observation and Action Models (LOAM), a framework leveraging LLMs to construct refined agent spaces from raw environments. To mitigate the computational burden of identifying the best candidate model from stochastic LLM outputs, LOAM incorporates a continuous racing mechanism that dynamically allocates resources to prioritize the most promising configurations without additional training overhead. Empirical evaluations on HumanoidBench and Isaac Lab demonstrate that LOAM consistently outperforms handcrafted baselines in both learning speed and asymptotic performance.

## 1. Introduction

Large Language Models (LLMs) have evolved beyond simple text generation into powerful tools for semantic reasoning. By synthesizing vast amounts of technical information, LLMs have demonstrated a remarkable ability to understand and model intricate systems, often rivaling or even surpassing human experts in specialized tasks (Luo et al., 2025). This capability allows for new ways to formalize and solve complex problems that were previously dependent on manual human intuition (Lu et al., 2024; Hong et al., 2025).

This paradigm extends to Reinforcement Learning (RL).

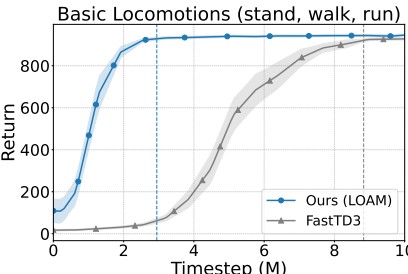

*Figure 1.* LOAM is 3× faster than default benchmark model (FastTD3) across HumanoidBench stand, walk, and run tasks.

While initial breakthroughs have primarily leveraged LLMs for reward design (Ma et al., 2024; Xie et al., 2024), a process traditionally constrained by laborious human trial-and-error, the equally critical task of designing the environment—including observation and action spaces—remains relatively underexplored (Park et al., 2024).

A major bottleneck in utilizing LLMs for environment design is the inherent stochasticity and variable quality of LLM-generated outputs. Since an inadequate agent space can lead to training failure, identifying the best configuration is crucial; however, the resulting selection process remains computationally prohibitive. Previous works (Ma et al., 2024; Xie et al., 2024; Wang et al., 2024) address this by generating and evaluating multiple candidates—often necessitating training each to convergence to identify the best-performing one—imposing a computational burden.

To address these challenges, we introduce LOAM (LLM-based design of Observation and Action Models). LOAM leverages the semantic reasoning of LLMs to construct refined agent spaces from raw environments, tailored to the task's requirements. Our experiments demonstrate that these LLM-designed spaces uncover structures that consistently outperform human-engineered baselines, achieving over 3× faster learning (Figure 1). Central to our framework is LOAM-Race, a continuous racing mechanism that dynamically allocates resources to prioritize the most promising configurations. Crucially, LOAM-Race identifies the best configuration within the same cumulative timestep budget as a single-model training run, effectively eliminating the efficiency bottlenecks of previous selection methods.

[1]LG AI Research, Seoul, Republic of Korea. Correspondence to: Kanghoon Lee <kanghoon.lee@lgresearch.ai>.

*Proceedings of the 43rd International Conference on Machine Learning*, Seoul, South Korea. PMLR 306, 2026. Copyright 2026 by the author(s).

This work makes three primary contributions:

1. We propose LOAM, a framework that leverages the semantic reasoning capabilities of LLMs to design task-relevant observation and action models in a structured form (i.e., Python code).

2. We introduce LOAM-Race that handles the inherent uncertainty of LLM by using predictive acquisition scores to find best configurations within a standard single-model budget.

3. Evaluations on HumanoidBench and Isaac Lab show that LOAM consistently outperforms handcrafted designs, significantly accelerating learning while maintaining or surpassing final performance.

## 2. Preliminary

### 2.1. RL Problem Formulation

In many sequential decision-making scenarios, agents lack access to the full underlying state of the environment due to sensor noise, occlusion, or inherent system complexity. Under this partial observability, we formulate the problem as a Partially Observable Markov Decision Process (POMDP).

A POMDP is formally defined by the tuple $\mathcal{M} = \langle \mathcal{S}, \mathcal{O}, \mathcal{A}, P, \Omega, R, \rho, \gamma \rangle$, where $\mathcal{S}$ is the state space, $\mathcal{O}$ the observation space, $\mathcal{A}$ the action space, $P(s'|s, a)$ the transition dynamics, $\Omega(s)$ the observation function, $R(s, a)$ the reward function, $\rho(s_0)$ the initial state distribution, and $\gamma$ the discount factor. The goal is to find a policy $\pi$ that maximizes the expected discounted return:

$$\pi^* = \arg\max_{\pi} \mathbb{E}\Big[ \sum_{t=0}^{\infty} \gamma^t R(s_t, a_t) \Big].$$

### 2.2. Decoupling Agent Space from Raw Environment

To address the gap between complex environment and efficient learning requirements, we formally distinguish between the raw environment spaces and the constructed spaces for the agent.

**Definition 2.1** (Raw Environment Space). The *Raw Space* represents the direct space provided by the underlying environment. It consists of:

- **Raw Observation Space ($\Omega_{\mathbf{raw}}$)**: The complete set of all accessible features provided by the environment.

- **Raw Action Space ($\mathcal{A}_{\mathbf{raw}}$)**: The comprehensive set of all controllable decision variables accepted by the environment.

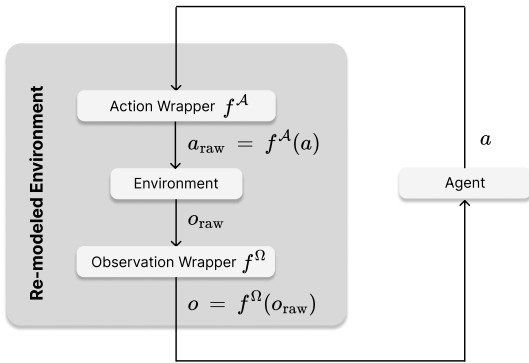

*Figure 2.* Re-modeled environment based on the observation modeling $f^\Omega$ and action modeling $f^\mathcal{A}$.

**Definition 2.2** (Agent Space). The *Agent Space*, denoted by the tuple $\langle \Omega, \mathcal{A} \rangle$, is defined as the refined spaces designed to facilitate the learning process of the policy $\pi$. Unlike the raw spaces, these spaces serve as the pre-requisite domain on which the policy is instantiated and optimized:

- **Agent Observation Space ($\Omega$)**: The refined feature set transformed from $\Omega_{\mathrm{raw}}$ for observation representation.

- **Agent Action Space ($\mathcal{A}$)**: The structured decision space mapped to $\mathcal{A}_{\mathrm{raw}}$ for efficient control.

Traditionally, constructing these agent spaces relies heavily on domain expertise and manual heuristics. This dependency limits the scalability of RL agents across diverse environments. In this work, we address this fundamental challenge by utilizing LLMs to design such agent spaces.

## 3. LOAM: LLM-based Design of Observation and Action Models

Instead of relying on handcrafted features, we leverage the semantic reasoning capabilities of LLMs to design observation and action spaces for the agent. Specifically, LOAM designs these spaces using task and environment information similar to what human designers rely on, substantially reducing the burden of manual feature engineering. In this section, we first formally define the observation and action modeling problem and explain how these models transform raw spaces into effective agent spaces. We then describe the structured prompts guiding the LLM to synthesize these models.

### 3.1. Observation and Action Modeling Problem

Standard RL research rarely operates directly on raw spaces due to their high dimensionality and lack of semantic abstraction. Therefore, bridging the gap between the raw spaces $\langle \Omega_{\mathrm{raw}}, \mathcal{A}_{\mathrm{raw}} \rangle$ and the agent spaces $\langle \Omega, \mathcal{A} \rangle$ is a critical design problem. We formulate this bridge as two mappings:

*Table 1.* Structural components of LOAM prompts and their modification requirements. The *System Prompt* provides shared context for both models, while the *Observation/Action Model Prompts* provide per-model generation instructions. Most components are reusable across tasks and domains; only *Task Description*, *Available Variable Info*, and *Action Mapping* require adaptation across environments.

| Prompt Type | Components | Description | Modification Requirement |
|---|---|---|---|
| System Prompt | Role and Objective | LLM persona and design objective | Not required |
| | Task Description | Task objective, initialization, and termination | Required when task changes |
| | Available Variable Info | Observation/action attributes and indices | Required when domain changes |
| Observation/Action Model Prompts | Instruction | Function signature and I/O specification | Not required |
| | Reasoning & Planning | Step-by-step task analysis guidelines | Not required |
| | Action Mapping (Action only) | Action dimension specifications | Required when domain changes |
| | Output Guide | Python function formatting guidelines | Not required |

- **Observation Modeling ($f^{\Omega} : \Omega_{\mathbf{raw}} \to \Omega$):** This mapping abstracts high-dimensional raw observations $o_{\mathrm{raw}} \in \Omega_{\mathrm{raw}}$ into succinct representations $o \in \Omega$ that capture salient features, thereby ensuring sample efficiency.

- **Action Modeling ($f^{\mathcal{A}} : \mathcal{A} \to \mathcal{A}_{\mathbf{raw}}$):** This mapping transforms actions $a \in \mathcal{A}$—which govern the granularity of decision-making—into raw decision variables $a_{\mathrm{raw}} \in \mathcal{A}_{\mathrm{raw}}$. Inadequate design here often yields prohibitive exploration complexity.

LOAM synthesizes these models $\langle f^{\Omega}, f^{\mathcal{A}} \rangle$ and combines them to transform the original environment into a *remodeled environment*. As shown in Figure 2, this wrapper effectively bridges the raw and agent spaces, allowing the policy to be trained seamlessly within the designed space.

### 3.2. Model Design Prompts

To guide the LLM, we structure the input context into three distinct categories: *System Prompt*, *Observation Model Prompt*, and *Action Model Prompt*.

**System Prompt** serves as a common context, establishing the LLM's role as an environment designer and providing shared information about the task and the environment's state representation. In contrast, the specific **Observation/Action Model Prompts** provide per-model function-generation instructions—covering the function signature, reasoning guidelines, and output format—with the Action Model Prompt additionally specifying the action space.

To construct the final query, the System Prompt is concatenated with the corresponding Model Prompt: it is paired with the Observation Model Prompt to synthesize the observation model $f^{\Omega}$, and with the Action Model Prompt to synthesize the action model $f^{\mathcal{A}}$.

Importantly, the Action Model Prompt does not impose a fixed latent action structure across tasks. Instead, it guides the LLM to determine the dimensionality and organization of the action space from the task description, and domain semantics, allowing the generated action model to reduce, preserve, or expand the dimensionality of action depending on the task requirements.

### 3.3. Prompt Reusability and Modification Requirements

Most prompt components are reusable across tasks and domains and do not require task-specific expert knowledge. In practice, only three components require adaptation when the environment changes: *Task Description*, *Available Variable Info*, and *Action Mapping*. The task description only specifies the objective, initialization, and termination conditions, while the variable information and action mapping are directly collected from environment documentation, APIs, or XML specifications. Thus, the required human effort is minimal and mainly involves organizing semantic information provided by the environment into our prompt format.

Table 1 presents the structural components of these prompts, and the required modification. For the full prompt templates, please refer to Appendix A.

## 4. LOAM-Race: Racing Multiple Designs via Dynamic Resource Allocation

Due to the inherent stochasticity of LLMs, generated observation or action spaces can occasionally be inadequate, leading to total training failure or performance significantly inferior to handcrafted baselines. To mitigate the risks associated with LLM quality diversity, it is essential to evaluate multiple candidate designs.

In this section, we first discuss why traditional selection methods are computationally prohibitive for practical RL applications. We then present LOAM-Race, a mechanism that overcomes these efficiency bottlenecks by strategically allocating training resources based on predictive acquisition scores.

*Figure 3.* Prior two-stage approaches generate multiple candidate models and train each for a substantial portion of the total budget $T$ (e.g., $0.8T$ in LESR) until partial convergence. The best performing candidate is then refined by an LLM to initialize the next iteration, repeating this iterative process to improve designs. In contrast, LOAM-Race bypasses these iterative procedure. By adaptively identifying and prioritizing high-potential models within a standard single-model training run, LOAM-Race achieves significantly higher computational efficiency while identifying best designs.

## 4.1. The Computational Burden of Model Selection

Prior works (Ma et al., 2024; Wang et al., 2024) address the inherent stochasticity of LLMs through two-phase strategy consisting of generating multiple candidate models and selecting the best-performing one. However, identifying the best candidate model imposes a significant computational burden, as it typically necessitates training each candidate policy to convergence to accurately assess its performance. For instance, LESR (Wang et al., 2024) evaluates 18 candidate models by training each for $0.8T$, where $T$ is the timestep budget for a single-model training. Following this, the best model undergoes a full training ($1T$). Consequently, this approach consumes $18 \times 0.8T + 1T = 15.4T$ in total—roughly 15 times the cost of training a policy on a single model—which substantially degrades sample efficiency.

## 4.2. Predictive Selection Strategy via Optimistic Return Extrapolation

We introduce LOAM-Race to address this limitation. The rationale behind LOAM-Race is that the future potential of a candidate model can be estimated without training its policy to convergence, enabling efficient model selection with significantly reduced computational cost. A naive selection criterion would simply choose the model with the highest current return. However, early-stage returns are noisy estimates of final performance, and a model that lags initially may eventually surpass others.

To account for this uncertainty, we adopt the principle of *Optimism in the Face of Uncertainty (OFU)* (Jamieson et al., 2014; Auer et al., 2002; Maron & Moore, 1993): instead of selecting based on current performance alone, we select the model with the highest upper confidence bound on predicted final performance. This acquisition function balances exploitation (favoring currently strong models) with exploration (giving uncertain models a chance to prove themselves). To compute these bounds, serving as the acqui-

---

**Algorithm 1** LOAM-Race
___
**Require:** $K$: number of environment models, $g$: acquisition function, $T$: maximum timesteps
**Ensure:** The best policy $\pi^*$ and environment model $\mathcal{M}^*$
 1: Generate $K$ environment models $\mathcal{M}_k, k = 1, \ldots K$ using LOAM
 2: Initialize return histories $\mathcal{R}_k, k = 1, \ldots K$
 3: $i \leftarrow 1$
 4: **while** total timesteps $< T$ **do**
 5: $\quad$ # *Allocate training resources for $i$-th model*
 6: $\quad$ Rollout $\pi_i$ on $\mathcal{M}_i$ for race timesteps and update returns to $\mathcal{R}_i$
 7: $\quad$ Train $\pi_i$ with collected experiences
 8: $\quad$ # *Select most promising model using acquisition score*
 9: $\quad i \leftarrow \arg\max_k g(\mathcal{R}_k)$
10: **end while**
11: # *Select the best performing policy and its environment model*
12: $i \leftarrow \arg\max_k \max(\mathcal{R}_k)$
13: **return** $\pi_i, \mathcal{M}_i$
___

sition score for each candidate model, we extrapolate each candidate model's return trajectory using *Bayesian ridge regression (BRR)*.

## 4.3. Overall Procedure

The LOAM-Race algorithm is outlined in Algorithm 1. Initially, all $K$ models are trained for a race timestep ($T/20$). This phase establishes baseline performance estimates and populates the return histories ($\mathcal{R}_k$) necessary for score calculation. Training resources are then allocated based on an acquisition score $g(\mathcal{R}_k)$ that balances current performance with future potential. Specifically, the BRR model uses training session indices as inputs and evaluation returns as targets, and estimates both the predicted future return and

its uncertainty within the remaining budget. The acquisition score is then computed as the predicted mean plus predicted standard deviation. In each iteration, the model with the highest score ($i = \mathrm{argmax}_k g(\mathcal{R}_k)$) is exclusively assigned the race timestep for training. Once the total budget $T$ is exhausted, LOAM-Race returns the policy and environment model that achieved the highest overall return: $\mathrm{argmax}_k \max(\mathcal{R}_k)$.

Figure 3 provides a schematic illustration of the two-stage procedure from prior works, and our dynamic resource allocation procedure, which enables the **same cumulative timestep budget** as a standard single-model run. While prior works (Ma et al., 2024; Xie et al., 2024; Wang et al., 2024) often employ iterative LLM-based refinement to improve candidate designs, our preliminary experiments indicated that such feedback loops yield minimal improvements if the initial design is fundamentally flawed. Consequently, we omit the refinement process to maintain overall efficiency; a detailed analysis are provided in Appendix F.4.

## 5. Related Work

Recent studies have increasingly leveraged LLMs to improve RL by designing key components of the learning environment. Much of this work has focused on reward design, where LLMs are used to overcome sparse or delayed feedback by automatically generating reward functions (Ma et al., 2024; Kwon et al., 2023; Xie et al., 2024; Field et al., 2025). EUREKA (Ma et al., 2024) employs an evolutionary search algorithm with reflection to iteratively optimize reward model (i.e., reward function code) based on policy training statistics. Similarly, Text2Reward (Xie et al., 2024) generates dense reward model from natural language goals using a compact environment abstraction and allows for iterative refinement via human feedback. While reward model designing has been the main emphasis, recent efforts have also begun to examine the role of observation and action models, exploring the automated specification (Chandak et al., 2019; Kim & Ha, 2021; Jia et al., 2025).

Within this direction, a recent position paper has emphasized the broader need for environment designing in RL, drawing attention to the particular difficulty of jointly designing observation and action spaces (Park et al., 2024). A closely related approach is LESR (Wang et al., 2024), which enhances RL efficiency by appending additional LLM-generated features to pre-existing, namely handcafted, observations and rewards. Unlike LESR, LOAM generates observation and action models from scratch directly from raw observation. As LLM outputs are inherently stochastic, previous works generate multiple candidate models, train and evaluate them through pilot experiments iteratively, and then train on the best-performing design before final evaluation. This two-stage procedure requires additional ex-

perimentation, whereas LOAM-Race identifies promising observation–action models within a single experiment using a racing strategy.

A complementary line of work studies representation learning and latent action abstraction for RL, which learns compact state or action representations through auxiliary objectives or structured latent spaces (Fujimoto et al., 2023; Bharadhwaj et al., 2022; Jiang et al., 2023; Allshire et al., 2021). These methods typically learn representations on top of a predefined environment interface, whereas LOAM addresses the orthogonal problem of using LLM semantic reasoning to design the observation and action models that define the interface itself.

## 6. Experiments

In this section, we empirically validate LOAM's capability to generate observation and action models enabling sample-efficient learning for diverse continuous control problems. Our evaluation is conducted on both HumanoidBench (Sferrazza et al., 2024) and NVIDIA Isaac Lab (Mittal et al., 2023) to ensure robustness and broad applicability. We first detail the experimental setup and main results, and conclude with ablation studies and qualitative evaluations.

### 6.1. Experimental Setup

We evaluate our method on two high-dimensional continuous control benchmarks; HumanoidBench and Isaac Lab. For both environments, we utilize FastTD3 as a backbone RL algorithm and OpenAI's GPT-5 (`gpt-5-2025-08-07`) as a backbone for all LLM-based algorithms. FastTD3 (Seo et al., 2025) is a high-performance variant of TD3 (Fujimoto et al., 2018) tailored for humanoid control; it incorporates parallel simulation, large-batch updates, and a distributional critic to accelerate and stabilize training.

For LOAM-Race, we use the number of candidate models ($K$) as 3 and the race timestep as 128K. We use $\Omega_{\mathrm{raw}}$ only for LOAM-variants because existing methods suffer from the curse of dimensionality. We report average returns and standard errors over 5 runs.

**HumanoidBench.** This environment features humanoid robots equipped with dexterous hands and includes a variety of challenging whole-body manipulation and locomotion tasks, as shown in Figure 4 and Table 2. We define the raw observation space $\Omega_{\mathrm{raw}}$ to consist solely of fundamental physical states directly accessible through the `mj.data` API, resulting in a high-dimensional space where $o_{\mathrm{raw}}$ exceeds 3,107 dimensions (see Appendix D.1 for details).

To ensure tractability, HumanoidBench provides a handcrafted observation space $\Omega_{\mathrm{hand}}$. This space primarily re-

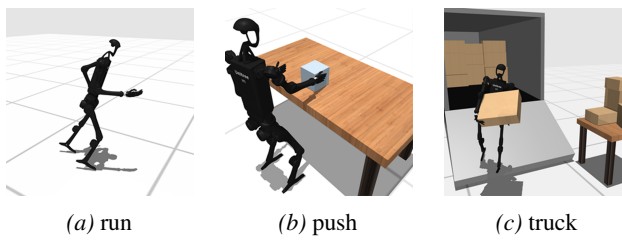

*(a)* run       *(b)* push       *(c)* truck

*Figure 4.* Examples of HumanoidBench tasks. We evaluate LOAM across three categories: (a) **Locomotion**: moving forward or navigating (e.g. run); (b) **Static manipulation**: performing dexterous manipulation while stationary (e.g. push); (c) **Dynamic manipulation**: performing manipulation while moving (e.g. truck).

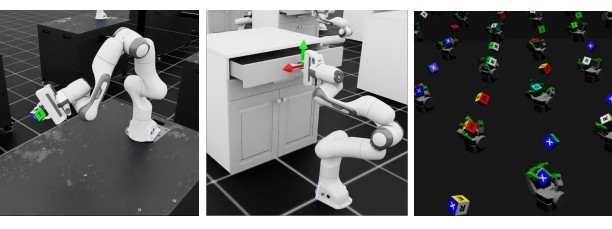

*(a)* Lift-Cube       *(b)* Open-Drawer       *(c)* Repose-Cube

*Figure 5.* Examples of Isaac Lab tasks. (a) Picking a cube to a target position. (b) Grasping and opening a drawer. (c) In-hand cube reorientation.

duces $\Omega_{\text{raw}}$ to joint positions and velocities, yielding a 151-dimensional vector (or larger in specific tasks [1]).

**Isaac Lab.** Unlike HumanoidBench, this environment presents distinct physics and morphological settings. We employ three tasks shown in Figure 5: Lift-Cube and Open-Drawer (single-arm manipulator), and Repose-Cube (dexterous hand). We align $\Omega_{\text{raw}}$ with the isaaclab.assets.RigidObjectData API, resulting in a high-dimensional space ($|o_{\text{raw}}| > 1, 100$)[2], whereas $\Omega_{\text{hand}}$ consists of approximately 30 dimensions. To adapt LOAM to Isaac Lab, we substitute domain-specific details from official specifications[3] into the corresponding components such as *Task Description* within our structured prompts (Table 1). We believe this modular design demonstrates that LOAM can be extended to other domains simply by providing the relevant domain-specific context.

### 6.2. Results on HumanoidBench

We evaluate LOAM against two baselines: FastTD3 (standard RL without LLM guidance) and LESR (LLM-

---

[1]For instance, in h1hand-reach-v0, the 3D positions of **the** left hand and target goal are additionally provided.

[2]We intentionally excluded provided highly-engineered features to necessitate learning relevant features directly from raw observations (see Appendix D.2).

[3]Please refer to https://isaac-sim.github.io/IsaacLab/main/index.html for details.

---

*Table 2.* HumanoidBench tasks categorized by required skill.

| Task | Description |
|------|-------------|
| ***Locomotion*** | |
| stand | Maintaining a standing pose |
| walk | Walking forward at a speed of 1m/s |
| run | Running forward at a speed of 5m/s |
| hurdle | Running forward while overcoming hurdles |
| slide | Walking forward over slides |
| reach | Reaching for a 3D point and touching with the left hand |
| ***Static Manipulation*** | |
| push | Moving a box on a table to a target point |
| cube | Manipulating two cubes in-hand to reach a target orientation |
| insert | Inserting a peg into two tight target blocks |
| ***Dynamic Manipulation*** | |
| truck | Unloading packages from a truck |
| powerlift | Lifting a barbell |
| package | Carrying a box to a target point |

augmented). While LESR[4] utilizes an augmented observation $[o_{\text{hand}}, f^{\Omega}(o_{\text{hand}})]$, where $o_{\text{hand}} \in \Omega_{\text{hand}}$, LOAM typically employs a direct, compact representation derived from raw observations: $f^{\Omega}(o_{\text{raw}})$. To ensure a fair comparison using the same underlying information, we also evaluate LOAM$_{\text{hand}}$, which applies the modeling process to the handcrafted observations (i.e., $f^{\Omega}(o_{\text{hand}})$).

As shown in Figure 6, LOAM reliably surpasses FastTD3 and LESR across all tasks excluding package, where all methods struggle. In tasks such as run, hurdle, slide, and push, LESR performance falls below FastTD3. This can be attributed to the high-dimensional feature space introduced by LESR, which is twice as large as the handcrafted observation. This highlights a fundamental design difference: while LESR simply appends features to potentially suboptimal handcrafted observations, LOAM directly constructs entirely new, compact, and task-relevant representations from the raw observation space.

While sample-efficient, LOAM can be unstable in tasks like insert_small and cube when the LLM generates uninformative features. LOAM-Race mitigates this stochasticity; despite a slight efficiency trade-off, it consistently achieves the highest asymptotic returns and superior robustness against the randomness of LLM-generated designs.

---

[4]Importantly, LESR requires approximately 4.5× more training samples due to nine pilot experiments for best selection. LESR runs nine pilot experiments (three iterations of generating three candidate models, each trained with half the total budget), after which the best observation design is used for the final report.

*(a)* Locomotion tasks

*(b)* Static manipulation tasks

*(c)* Dynamic manipulation tasks

*Figure 6.* Learning curves on 12 HumanoidBench tasks. We report the average return over 5 random seeds, with shaded regions indicating standard error.

*Figure 7.* Learning curves on 3 Isaac Lab tasks. We report the average return over 5 random seeds, with shaded regions indicating standard error.

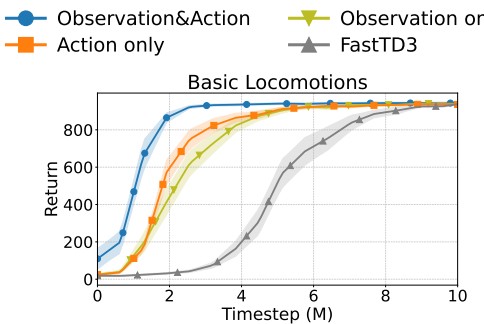

Figure 8. Ablation study on observation and action models.

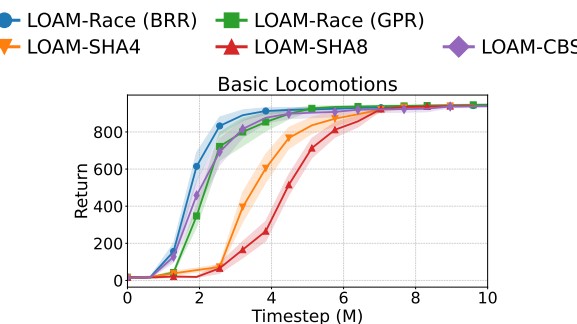

Figure 9. Comparison of LOAM-Race with other candidate selection strategies.

## 6.3. Results on Isaac Lab

We evaluate LOAM on Isaac Lab against FastTD3 and LESR to test generalization beyond Humanoid-Bench. As shown in Figure 7, LOAM and LOAM-Race consistently outperform FastTD3 in both convergence speed and final performance. In the Franka tasks (`Lift-Cube`, `Open-Drawer`), LOAM learns substantially faster than the baselines, while in the high-dimensional `Repose-Cube` task, LOAM-Race achieves the best asymptotic performance despite a slight early-training delay from the racing mechanism.

LESR also improves over FastTD3 by augmenting hand-crafted observations with LLM-generated features. In contrast, LOAM-Race directly constructs effective observation and action spaces from semantic environment information without relying on handcrafted designs, while still achieving competitive or superior performance. Overall, these results demonstrate strong generalization across different simulation environments and robot embodiments.

A qualitative analysis of the LOAM-designed models, including the complete code and line-by-line descriptions, can be found in Appendix B.

## 6.4. Further Analysis

**Effect of observation and action models.** We compare the full LOAM framework against observation-only and action-only variants on locomotion tasks, with results shown in Figure 8. Both variants improve learning efficiency over the baseline, indicating that observation and action model are each individually beneficial. However, the full LOAM framework consistently achieves the fastest convergence, demonstrating that jointly optimizing both observation and action spaces provides the greatest performance gain.

**Effect of the candidate selection strategy.** To analyze the impact of the acquisition function and resource allocation strategy in LOAM-Race, we compare our default Bayesian Ridge Regression (BRR)-based racing approach against several alternatives: LOAM-Race with Gaussian Pro-

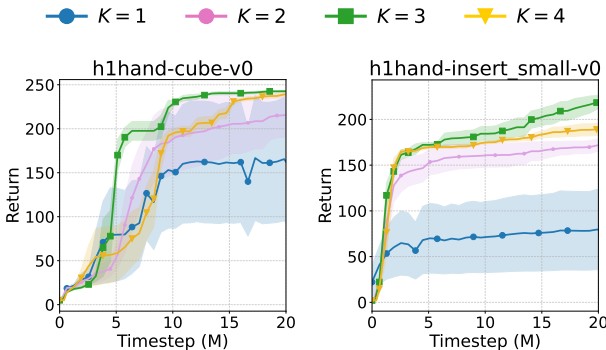

Figure 10. Impact of the number of candidate models ($K$) in LOAM-Race.

cess Regression (GPR); LOAM-SHA4 and LOAM-SHA8, Successive Halving (Jamieson & Talwalkar, 2016)-based methods with 4 and 8 candidates; and LOAM-CBS, which always selects the current best candidate.

As shown in Figure 9, BRR achieves the fastest and most stable convergence, while GPR offers little practical advantage despite its higher complexity. Among resource allocation strategies, the racing-based approaches achieve the best overall performance, whereas the SHA-based methods converge more slowly in our setting. Overall, these results suggest that LOAM-Race effectively balances exploration and exploitation for robust candidate selection (See details in Appendix F.2).

**Effect of the number of candidate models in LOAM-Race.** Figure 10 illustrates the trade-off between robustness and convergence speed. With more candidate models, performance variance narrows, consistent with the intuition that a larger pool is more likely to include at least one strong candidate model and thereby improves robustness. However, once a strong candidate model is present, additional ones merely dilute the training budget, slowing convergence, as seen in $K = 4$ being slower than $K = 3$. From this ablation, we find that $K = 3$ provides a favorable balance between robustness and efficiency.

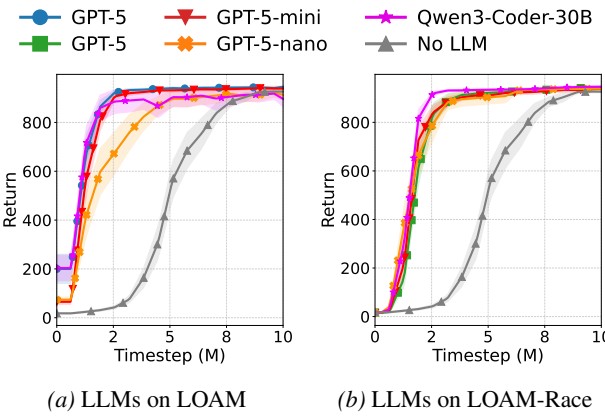

*(a)* LLMs on LOAM   *(b)* LLMs on LOAM-Race

*Figure 11.* Comparison of different LLMs including GPT-5 variants, Qwen3-Coder-30B, and a non-LLM baseline (FastTD3) on HumanoidBench basic locomotion tasks.

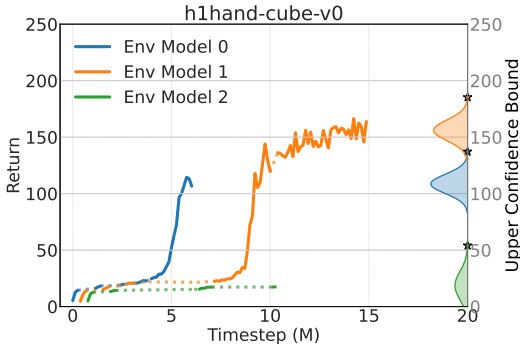

*Figure 12.* Visualization of LOAM-Race. The left and right y-axes denote the model's average return and the acquisition score.

**Effect of the LLM backbone.** Our ablation study comparing the standard GPT-5 backbone against lighter and open-weights variants—GPT-5-mini, GPT-5-nano, and Qwen3-Coder-30B-A3B-Instruct (Yang et al., 2025)—reveals that while standard LOAM exhibits performance degradation with reduced model capacity (particularly in 30B and nano models) (Figure 11a), LOAM-Race effectively mitigates this sensitivity (Figure 11b). Notably, LOAM-Race enables these smaller models to achieve results comparable to the GPT-5, confirming that our racing mechanism unlocks the potential of cost-effective backbones and reduces dependency on proprietary systems (See details in Appendix F.3).

**Visualization of LOAM-Race.** We visualize the racing procedure of LOAM-Race during training in Figure 12. Every race timesteps (128K timesteps), LOAM-Race estimates the upper confidence bounds of all $K$ candidate environment models and selects one. The following race timesteps are used to update the policy associated with the selected models, and subsequently the corresponding upper confidence bound is further refined. In this figure, LOAM-Race concentrates resources on promising candidate models (models 0 and 1) while, after minimal exploration, allocating essentially none to the weak candidate model (model 2).

## 7. Conclusion

We introduced LOAM, a framework leveraging LLM semantic reasoning to design structured observation and action models. To mitigate the stochasticity of LLM designs, we proposed LOAM-Race, which identifies the most promising configurations within a standard single-model budget using predictive acquisition scores. Our experiments show that not only substantially reduces the need for manual feature engineering but also outperforms human-designed models across all HumanoidBench tasks. By shifting the complexity of observation and action design to a LLM-based frame-

work, we alleviate a critical bottleneck that has limited RL deployment in complex systems.

## Impact Statement

This paper presents work whose goal is to advance the field of machine learning by facilitating the design of observation and action models for RL via LLMs. By replacing handcrafted specifications with LLM-generated functions, our framework, LOAM, demonstrates a significant reduction in the manual engineering effort required for complex control tasks. While the LLM-driven design and use of RL observation and action spaces carry potential societal and ethical implications, we believe these are consistent with the broader challenges and safety standards well-established within the field of machine learning. Therefore, we consider that the expected societal consequences of our work are aligned with the general advancements in system design and do not require further specific highlighting.

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

## A. Full Prompt Templates

This section details the structured full prompts that drive the LOAM framework. Our methodology is a modular pipeline that begins with a **System Prompt** to provide task-specific context and define the LLM's expert persona. It then deploys its core modules for **Observation Model Prompt**, which generates a compact and informative state representation for the agent, and **Action Model Prompt**, which structures a coordinated and efficient control interface for the robot's actuators.

To ground these prompts in a concrete example, the templates shown are tailored for the H1Hand robot. The H1Hand morphology consists of a floating pelvis, two five-DoF legs, a single-DoF torso joint, two five-DoF arms, and two dexterous five-digit hands. The simulator exports 76 generalized positions (`qpos`) and 75 generalized velocities (`qvel`). This detailed structural information, including index mappings for joints and sensors, is provided to the LLM within the System Prompt to ensure any generated code can address the correct components.

Figure 13 provides a visual overview of the stages involved in this pipeline.

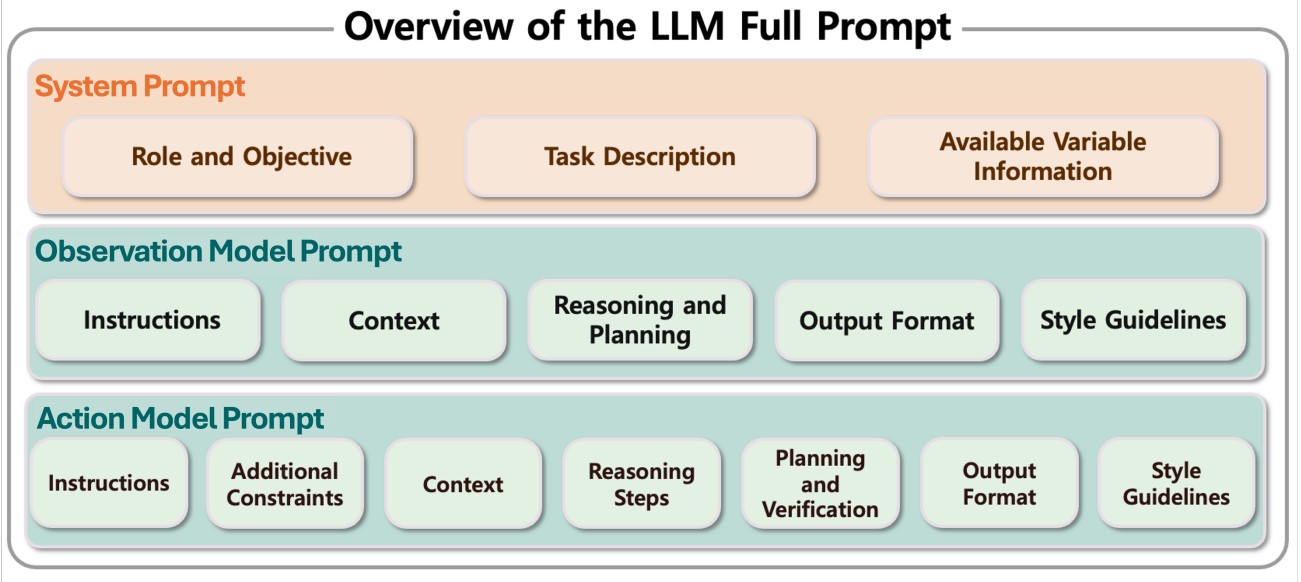

*Figure 13.* Overview of the LLM Full Prompt Template. The template is organized into three main components. System Prompt establishes role and objective, task description, and available variable information of the LLM. Observation Model Prompt details the instructions, context, reasoning, output format, and style for generating the state representation function. Action Model Prompt provides a similar breakdown for synthesizing the action space, including additional constraints, reasoning steps for design, and verification plans.

### A.1. Prompt Structure

Our framework is built upon a structured suite of prompts, where each module serves a distinct function in the code generation pipeline. As illustrated in Figure 13, this suite is composed of three primary modules. Below, we detail the full template for each component.

- **System Prompt:** The pipeline begins with the System Prompt, which grounds the LLM in the global context. it serves three primary functions: defining the LLM's expert persona **Role and Objective**, providing the complete operational details of the task **Task Description** (Context), and detailing all available simulator state variables **Available Variable Information** (Data attributes). This initial stage ensures the LLM is fully contextualized before any code generation begins.

> **System Prompt**
>
> ```
> # Role and Objective
> You are an expert robotics engineer and AI researcher specializing in Deep
> Reinforcement Learning. Your objective is to design an optimal, {TYPES} for a
> reinforcement learning agent, enabling efficient and effective learning for the
> specified task.
>
> # Context
> - Task description :
> {TASK_DESCRIPTION}
> - The `data` object holds simulation states; all attributes are torch tensors with
>  a batch dimension as the first axis.
> - Detailed information about the "data" object's attributes can be found in the
> below "Detailed data attributes" section, which describes what each index in the
> data attribute represents.
>
> # Data attributes
> {AVAILABLE_STATE_INFO}
> ```

- **Observation Model Prompt:** This module guides the synthesis of a self-contained Python function, `compute_obs`. The prompt is structured with five key components. It provides explicit **Instructions** on the function's requirements and defines its operational **Context**, such as its signature. A **Reasoning and Planning** section prompts the LLM to select and engineer relevant features. Finally, the prompt enforces a specific **Output Format** (a single code block) and sets **Style Guidelines** (Verbosity), such as code verbosity and the use of vectorized operations, to ensure the generated code is efficient and robust for RL training.

> **Observation Model Prompt**
>
> ```
> # Instructions
> - Begin with a concise checklist (3-7 bullets) of what you will do; keep items
> conceptual, not implementation-level.
> - **Generate a complete Python function** named `compute_obs` that computes **and
> returns** the observation vector named `obs` (as a torch tensor).
> - The function signature **must be** `compute_obs(data)`. It should accept the
> simulation state object `data` as its sole argument.
> - The function **must return** the final `obs` tensor.
> - Inside the function, use only the explicitly allowed state variables from the `
> data` object. However, never use cfrc_ext data for any external, non-robot objects
> .
> - If you need a value not in the allowed variables, hard-code it as a constant
> per the Robot Specification-never try to access `data` or `self` for unspecified
> attributes.
> - The function's logic must correctly handle the batch dimension and be fully
> vectorized.
> - Any new tensors created inside the function must use the same device as
> existing tensors (e.g., `device=data.qpos.device`).
> - The output must be a single, raw Python code block labeled `python`, containing
> the entire function. Do not include any other text or explanations outside the
> code block.
>
> # Context
> - **Function Signature:** `def compute_obs(data):`
> - **Inputs:** The function will receive `data` (simulation state and sensor data).
>
> # Reasoning and Planning
> - **Plan the complete function structure**, including the signature `compute_obs(
> data)`, the internal logic for vector computation, and the final `return obs`
> ```

```
statement.
- Internally: Analyze variables and index mappings. Implement device-safe, batch-
capable extraction and concatenation with the correct shapes.
- **Considering the task description carefully**, select the most relevant
features (joint positions/velocities, site positions, target position) for the
task. If necessary, leverage your domain expertise in robotics and physics to
compute derived, physically meaningful features.
- For any physical task, maintaining the robot's stable and balanced posture is a
 fundamental and implicit requirement. The final code must reflect a deep
understanding of the need for whole-body postural control while performing the
primary task.
- After generating the function, validate that the returned observation vector
uses only allowed variables and that batch/device compatibility is preserved. If
validation fails, correct and re-verify.

# Output Format
- Output a single markdown `python` code block.
- The code block must begin with a detailed comment explaining the observation
design and structure.
- The code block must contain the **entire, self-contained `compute_obs` function
**, from the `def` statement to the final `return` statement.
- Do not include any inline/trailing comments or any text outside the code block.

# Verbosity
- Write code with high clarity and a readable structure. Be verbose in comments
at the top of the code block only.

Write a **complete and self-contained Python function** named `compute_obs` that
computes and returns the effective observation vector for efficient training of a
 deep reinforcement learning agent in this environment.
```

- **Action Model Prompt:** This prompt addresses the high-dimensional control challenge by instructing the LLM to design an *Action Expansion*: a deterministic mapping from a low-dimensional latent action to the full 61-actuator command vector. This mapping is typically achieved via predefined synergies—coordinated patterns where a single latent command drives multiple joints—and stabilizing PD terms. The prompt requires the LLM to first reason about an optimal latent action space based on the task and robot morphology.

To structure this process, the prompt provides detailed **Instructions**, defines the function **Context** including the full actuator map, and specifies **Additional Constraints** such as shape validation. It then guides the LLM through **Reasoning Steps** for synergy design and a **Planning and Verification** stage to ensure a correct mapping to the full actuator space. A strict **Output Format** and specific **Style Guidelines** are enforced to produce an efficient `compute_action` function that embeds strong domain priors, thereby constraining the agent's exploration problem.

---

**Action Model Prompt**

```
# Instructions
- Begin with a concise checklist (3-7 bullets) of what you will do.
- **First, determine and state an optimal, dimension for the input action space (`
action_dim`)** based on the actuator map and robotics principles.
- **Generate a complete Python function** with the signature `compute_action(
action, data)`.
- The function must accept a `action` tensor and the `data` object as inputs.
- The function must transform the `action` tensor (with the `action_dim` you
determined) into a full 61-dimensional control vector.
- The function **must return** the final 61-dimensional tensor, named `action_out
`.
- **Your output must follow the two-part format specified in the "Output Format"
section below.**
```

```
- The Python code block must start with a high-level comment summarizing your
chosen `action_dim` and mapping approach.
- All implementation must accommodate batch processing and handle device
consistency correctly.

## Sub-categories
- All tensor shapes must be validated to prevent mismatches.
- Any new tensors must be created on the same device as the input tensor `action`.
- Do not rename the function name or its parameters.

# Context
- **Function Signature:** `def compute_action(action, data):`
- **Inputs:**
    - `action` (torch.Tensor): **The input action tensor is the direct output from
 a policy neural network. Its values are typically in the range [-1, 1],
resulting from an activation function like `tanh`.**
    - `data` (simulation state and sensor data)
- Actuator mapping with 61 total outputs detailed below

# Actuator mapping (nu=61):
The final 61-dimensional output vector corresponds to the following actuators. **
Use this detailed map to decide on an appropriate input dimension and how it
should be expanded.**
Part 1: Body and Limbs (Indices 0-20)
# Legs (0-9)
[0] left_hip_yaw: Left hip's side-to-side rotation (Yaw)
[1] left_hip_roll: Left hip's side-to-side tilt (Roll)
[2] left_hip_pitch: Left hip's front-to-back movement (Pitch)
[3] left_knee: Left knee bending
[4] left_ankle: Left ankle movement
[5] right_hip_yaw: Right hip's side-to-side rotation (Yaw)
[6] right_hip_roll: Right hip's side-to-side tilt (Roll)
[7] right_hip_pitch: Right hip's front-to-back movement (Pitch)
[8] right_knee: Right knee bending
[9] right_ankle: Right ankle movement
# Torso (10)
[10] torso: Torso side-to-side rotation
# Arms (11-20)
[11] left_shoulder_pitch: Left shoulder's front-to-back movement (Pitch)
[12] left_shoulder_roll: Left shoulder's up-and-down movement (Roll)
[13] left_shoulder_yaw: Left shoulder's side-to-side rotation (Yaw)
[14] left_elbow: Left elbow bending
[15] left_wrist_yaw: Left wrist's side-to-side rotation (Yaw)
[16] right_shoulder_pitch: Right shoulder's front-to-back movement (Pitch)
[17] right_shoulder_roll: Right shoulder's up-and-down movement (Roll)
[18] right_shoulder_yaw: Right shoulder's side-to-side rotation (Yaw)
[19] right_elbow: Right elbow bending
[20] right_wrist_yaw: Right wrist's side-to-side rotation (Yaw)
Part 2: Left Hand (Indices 21-40)
# Wrist (21-22)
[21] lh_A_WRJ2: Left wrist pitch (up/down movement)
[22] lh_A_WRJ1: Left wrist roll (side-to-side tilt)
# Thumb (23-27)
[23] lh_A_THJ5: Left thumb base joint rotation
[24] lh_A_THJ4: Left thumb proximal joint (first knuckle) bend
[25] lh_A_THJ3: Left thumb hub joint
[26] lh_A_THJ2: Left thumb middle joint
[27] lh_A_THJ1: Left thumb distal joint (tip)
# Fingers (28-40)
[28] lh_A_FFJ4: Left index finger knuckle (side-to-side)
[29] lh_A_FFJ3: Left index finger proximal bend
[30] lh_A_FFJ0: Left index finger middle/distal bend (tendon)
```

```
[31] lh_A_MFJ4: Left middle finger knuckle (side-to-side)
[32] lh_A_MFJ3: Left middle finger proximal bend
[33] lh_A_MFJ0: Left middle finger middle/distal bend (tendon)
[34] lh_A_RFJ4: Left ring finger knuckle (side-to-side)
[35] lh_A_RFJ3: Left ring finger proximal bend
[36] lh_A_RFJ0: Left ring finger middle/distal bend (tendon)
[37] lh_A_LFJ5: Left little finger metacarpal joint
[38] lh_A_LFJ4: Left little finger knuckle (side-to-side)
[39] lh_A_LFJ3: Left little finger proximal bend
[40] lh_A_LFJ0: Left little finger middle/distal bend (tendon)
Part 3: Right Hand (Indices 41-60)
# This part mirrors the left hand's structure
# Wrist (41-42), Thumb (43-47), Fingers (48-60)
[41] rh_A_WRJ2: Right wrist pitch (up/down movement)
...
[60] rh_A_LFJ0: Right little finger middle/distal bend (tendon)

# Reasoning Steps
- **Plan the complete function structure**, including the signature `
compute_action(action, data)`, the internal mapping logic, and the final `return
action_out` statement.
- **Propose a suitable `action_dim`. ** Justify this choice in the code's summary
comment.
- **Considering the task description carefully**, transform the action tensor
intelligently to the full 61-dimensional space to create a smooth control
landscape for the neural network.
- For any physical task, maintaining the robot's stable and balanced posture is a
 fundamental and implicit requirement. The final code must reflect a deep
understanding of the need for whole-body postural control while performing the
primary task.
- Ensure correct expansion respecting batch and device.

# Planning and Verification
- Carefully decompose the expansion from **your chosen `action_dim`** to 61 using
the context and actuator map.
- Verify shape compatibility for all operations and ensure tensor device
consistency.
- After generating `action_out` but before returning it, include in-code
validation to confirm the output's batch and dimensional shape.

# Output Format
- **Your output must consist of two distinct parts, in this specific order. Do
not include any other text or explanation.**
- **Part 1: Action Dimension**
    - First, output the identifier `dimension`, followed by a newline, and then
the integer value for the chosen action dimension.
    - Example:
    ```
    dimension
    20
    ```
- **Part 2: Python Code**
    - Following the dimension, output a single Python code block with the
language identifier `python`.
    - This block must contain the **entire, self-contained `compute_action`
function**, from the `def` statement to the final `return` statement.

# Verbosity
- Code must include clear structure. Be verbose in the summary comment at the top
 of the code block only.

First, determine and output an optimal action space dimension. Then, write a **
```

```
complete and self-contained Python function** named `compute_action` that expands
an action tensor from this dimension into the full 61-dimensional control vector
and **returns** the result.
"""
```

## B. Qualitative Analysis of LOAM-Designed Observation and Action Models

In this section, we provide a qualitative analysis of the LOAM-designed observation and action models. We use the `h1hand-run-v0` locomotion task as an in-depth case study. The analysis first presents the fully populated prompt provided to the LLM for this task. It then details the baseline model, followed by the full code generated by our framework. Finally, we examine the generated design to identify the specific choices and structures that likely contribute to its effective performance. Additionally, we examine the action model generated for the `Isaac-Lift-Cube-Franka-v0` manipulation task to illustrate that LOAM does not simply compress original action spaces, but can also expand and restructure them depending on task requirements.

### B.1. Example of a Populated Prompt for `h1hand-run-v0`

The placeholders within the `System Prompt` templates shown in Section A, such as `{TYPES}`, `{TASK_DESCRIPTION}`, and `{AVAILABLE_STATE_INFO}`, are dynamically populated with the specifics of each target environment. This process grounds the LLM in the precise context of the task before code generation. For any given generation step, the system prompt templates are populated and concatenated following each specific tasks. For instance, to generate the designs for `h1hand-run-v0`, the system first populates the `System Prompt` template with the task's specific details and then appends the two universal templates for `Observation Model Prompt` and `Action Model Prompt` that are detailed in Section A. As a concrete example, the populated System Prompt for `h1hand-run-v0` is shown below.

---

**Example: System Prompt for `h1hand-run-v0`**

```
# --- Start of Populated System Prompt (from System Prompt Template) ---
# Role and Objective
You are an expert robotics engineer and AI researcher specializing in Deep
Reinforcement Learning. Your objective is to design an optimal, observation function
code, for a reinforcement learning agent, enabling efficient and effective learning
for the specified task.

# Context
- Task description :
Name:  h1hand-run-v0
Objective:  Keep forward velocity close to 5 m/s without falling to the ground.
Initialization:  The robot is initialized to a standing position, with random noise
added to all joint positions during each episode reset.
Termination:  The episode terminates after 1000 steps, or when z_pelvis < 0.2.

- The `data` object holds simulation states; all attributes are torch tensors with a
batch dimension as the first axis.
- Detailed information about the "data" object's attributes can be found in the below
 " Detailed data attributes" section, which describes what each index in the data
attribute represents.

# Data attributes
data.qpos
 - Shape:  (batch, nq=76)
 - Description:  Generalized positions (joint angles, base pose).
data.qvel
 - Shape:  (batch, nv=75)
 - Description:  Generalized velocities (joint rates, base velocity).
data.site_xpos
 - Shape:  (batch,nsite=9, 3)
 - Description:  3D position of each defined site (useful for end-effectors, sensors,
etc.).
data.qfrc_actuator
 - Shape:  (batch, nv=75)
 - Description:  Actuator forces mapped to degrees of freedom.
data.xpos
```

---

```
 – Shape:  (batch, nbody=71, 3)
 – Description:  3D position of each body.
data.xquat
 Shape:  (batch, nbody=71, 4)
 Description:  4D orientation (quaternion) of each body.
data.cvel
 Shape:  (batch, nbody=71, 6)
 Description:  6D spatial velocity (linear & angular) of each body.
data.cfrc_ext
 Shape:  (batch, nbody=71, 6)
 Description:  6D external wrench (force & torque) on each body.
data.cinert
 Shape:  (batch,nbody=71, 10)
 Description:  10D composite rigid body inertia of each body.
data.actuator_force
 Shape:  (batch, nu=61)
 Description:  Force/torque generated by each actuator.
data.sensordata
 Shape:  (batch, nsensor=14)
 Description:  Scalar sensor outputs.

Index Mapping for Key Arrays
This section provides the name-to-index mapping for the primary arrays.

data.qpos (nq=76, Generalized Positions)
[0:7] free_base:  Pelvis pose ([x, y, z, qw, qx, qy, qz])
[7:12] left_leg:  [hip_yaw, hip_roll, hip_pitch, knee, ankle]
[12:17] right_leg:  [hip_yaw, hip_roll, hip_pitch, knee, ankle]
[17] torso
[18:23] left_arm:  [shoulder_pitch, shoulder_roll, shoulder_yaw, elbow, wrist_yaw]
[23:47] left_hand:  24 joints
 – [23:25] Wrist (2j)
 – [25:29] Index Finger (4j)
 – [29:33] Middle Finger (4j)
 – [33:37] Ring Finger (4j)
 – [37:42] Little Finger (5j)
 – [42:47] Thumb (5j)
[47:52] right_arm:  [shoulder_pitch, shoulder_roll, shoulder_yaw, elbow, wrist_yaw]
[52:76] right_hand:  24 joints
 – [52:54] Wrist (2j)
 – [54:58] Index Finger (4j)
 – [58:62] Middle Finger (4j)
 – [62:66] Ring Finger (4j)
 – [66:71] Little Finger (5j)
 – [71:76] Thumb (5j)

data.qvel (nv=75, Generalized Velocities)
[0:6] free_base:  Pelvis velocity ([vx, vy, vz, wx, wy, wz])
[6:11] left_leg:  5 joint velocities
[11:16] right_leg:  5 joint velocities
[16] torso:  1 joint velocity
[17:22] left_arm:  5 joint velocities
[22:46] left_hand:  24 joint velocities (same structure as qpos)
[46:51] right_arm:  5 joint velocities
[51:75] right_hand:  24 joint velocities (same structure as qpos)

data.site_xpos (nsite=9, Site Positions)
[0] com
[1] left_foot
[2] right_foot
```

```
[3] left_eye
[4] right_eye
[5] head
[6] imu
[7] left_hand
[8] right_hand

data.xpos, data.xquat, data.cvel, data.cfrc_ext, data.cinert (nbody=71, Body-related
Arrays)
[0] world
[1] pelvis
[2:7] left_leg_links:  [hip_yaw, hip_roll, hip_pitch, knee, ankle]
[7:12] right_leg_links:  [hip_yaw, hip_roll, hip_pitch, knee, ankle]
[12] torso_link
[13:17] left_arm_links:  [shoulder_pitch, shoulder_roll, shoulder_yaw, elbow]
[17:42] left_hand:  25 bodies
 [17:20] Left Palm area (left_hand, wrist, palm)
 [20:24] Left Index Finger (knuckle, proximal, middle, distal)
 [24:28] Left Middle Finger (4 bodies)
 [28:32] Left Ring Finger (4 bodies)
 [32:37] Left Little Finger (5 bodies)
 [37:42] Left Thumb (5 bodies)
[46:71] right_hand:  25 bodies (same structure as left_hand)

data.actuator_force (nu=61, Actuator Forces)
[0:5] left_leg:  5 actuators
[5:10] right_leg:  5 actuators
[10] torso:  1 actuator
[11:16] left_arm:  5 actuators
[16:21] right_arm:  5 actuators
[21:41] left_hand:  20 actuators
 [21:23] Left Wrist (2a)
 [23:28] Left Thumb (5a)
 [28:31] Left Index Finger (3a)
 [31:34] Left Middle Finger (3a)
 [34:37] Left Ring Finger (3a)
 [37:41] Left Little Finger (4a)
[41:61] right_hand:  20 actuators
 [41:43] Right Wrist (2a)
 [43:48] Right Thumb (5a)
 [48:51] Right Index Finger (3a)
 [51:54] Right Middle Finger (3a)
 [54:57] Right Ring Finger (3a)
 [57:61] Right Little Finger (4a)

data.sensordata (nsensor=14, Sensor Outputs)
[0] left_foot_sensor (1D touch)
[1] right_foot_sensor (1D touch)
[2:5] pelvis_subtreelinvel (3D velocity)
[5:8] left_hand_subtreelinvel (3D velocity)
[8:11] right_hand_subtreelinvel (3D velocity)
[11:14] body_velocimeter (3D velocity)
```

**Observation Model Prompt**

The content of this prompt is identical to the "Observation Model Prompt" template shown in Section A.1.

**Action Model Prompt**

The content of this prompt is identical to the "Action Model Prompt" template shown in Section A.1.

### B.2. Case Study 1: `h1hand-run-v0` (Successful Locomotion)

The objective of this task is to command the H1 humanoid robot to run forward at a target velocity of 5.0 m/s while maintaining balance. This task primarily tests high-speed locomotion, dynamic stability, and rhythmic coordination.

#### B.2.1. BASELINE MODEL ANALYSIS

The baseline model utilizes a standard configuration where the observation and action spaces are high-dimensional and minimally processed.

- **Observation Space (151 dimensions):** The observation vector is formed by the direct concatenation of the robot's generalized positions (`qpos`, 76 dimensions) and velocities (`qvel`, 75 dimensions). This representation is comprehensive but includes task-irrelevant information (e.g., finger joint states) and is sensitive to the agent's global heading.

- **Action Space (61 dimensions):** The policy directly outputs a 61-dimensional vector, where each dimension corresponds to one of the robot's actuators. This provides full control but forces the agent to learn all inter-joint coordinations from scratch.

#### B.2.2. FULL LOAM-DESIGNED MODELS CODE FOR `h1hand-run-v0`

**LOAM-Designed Observation Model for `h1hand-run-v0`**

```
1  # Observation design for h1hand-run-v0 (final, aligned with 13D action)
2  # Goal: Run forward at ~5 m/s without falling. Observations emphasize base state
       in a heading-invariant manner,
3  # task goal, contacts, feet geometry, and only the joint subset that the policy
       controls (or influences via synergy).
4  #
5  # Structure (all batched torch tensors):
6  # 1) Base height:
7  #    - pelvis_z: qpos[:, 2] (1)
8  # 2) Base orientation/context (yaw invariant):
9  #    - gravity_in_base (3): world gravity [0,0,-1] expressed in pelvis frame via
       quaternion rotation
10 #    - heading_xy (2): pelvis forward (+X in pelvis) projected to world XY,
       normalized -> [cos(yaw), sin(yaw)]
11 # 3) Base velocities (in pelvis frame):
12 #    - v_body (3): linear velocity from sensordata (body_velocimeter)
13 #    - w_body (3): base angular velocity (qvel[:,3:6]) rotated into pelvis frame
14 # 4) Task-specific signals:
15 #    - speed_target (1): constant 5.0 m/s
16 #    - speed_error (1): v_body[:, 0] - 5.0 (forward speed error in pelvis frame)
17 # 5) Contacts:
```

```
18  #       - foot_contact (2): [left_foot_sensor, right_foot_sensor]
19  # 6) Geometry:
20  #       - feet_pos_in_base (6): left/right foot site positions relative to pelvis,
        expressed in pelvis frame
21  # 7) Controlled joint subset states (positions and velocities):
22  #       - pos (13): [L hip_roll, L hip_pitch, L knee, L ankle,
23  #                    R hip_roll, R hip_pitch, R knee, R ankle,
24  #                    torso,
25  #                    L shoulder_pitch, L elbow,
26  #                    R shoulder_pitch, R elbow]
27  #       - vel (13): same ordering as pos
28  #
29  # Total observation dimension: 1 + 3 + 2 + 3 + 3 + 1 + 1 + 2 + 6 + 13 + 13 = 48
30  def compute_obs(data):
31      import torch
32
33      device = data.qpos.device
34      dtype = data.qpos.dtype
35
36      # Quaternion helpers (q = [w, x, y, z])
37      def quat_normalize(q):
38          return q / torch.clamp(torch.linalg.norm(q, dim=-1, keepdim=True), min=1e
        -8)
39
40      def quat_conj(q):
41          qc = q.clone()
42          qc[..., 1:] = -qc[..., 1:]
43          return qc
44
45      # Rotate vector(s) v by quaternion q (unit). Supports v shape (B,3) or (B,N
        ,3).
46      def quat_rotate(q, v):
47          if v.dim() == q.dim() + 1:
48              q = q.unsqueeze(-2).expand(*v.shape[:-1], 4)
49          q = quat_normalize(q)
50          qvec = q[..., 1:]        # (..., 3)
51          qw = q[..., :1]          # (..., 1)
52          t = 2.0 * torch.cross(qvec, v, dim=-1)
53          return v + qw * t + torch.cross(qvec, t, dim=-1)
54
55      B = data.qpos.shape[0]
56
57      # Base pose
58      base_pos = data.qpos[:, 0:3]                    # (B,3) world
59      base_quat = quat_normalize(data.qpos[:, 3:7])   # (B,4)
60      base_quat_conj = quat_conj(base_quat)
61
62      # 1) Base height
63      pelvis_z = base_pos[:, 2:3] # (B,1)
64
65      # 2) Base orientation/context
66      g_world = torch.tensor([0.0, 0.0, -1.0], device=device, dtype=dtype).expand(B
        , 3)
67      gravity_in_base = quat_rotate(base_quat_conj, g_world)  # (B,3)
68
69      ex_body = torch.tensor([1.0, 0.0, 0.0], device=device, dtype=dtype).expand(B,
         3)
70      fwd_world = quat_rotate(base_quat, ex_body)  # (B,3)
71      heading_xy = fwd_world[:, :2]
72      heading_norm = torch.clamp(torch.linalg.norm(heading_xy, dim=-1, keepdim=True
        ), min=1e-8)
73      heading_xy = heading_xy / heading_norm  # (B,2)
```

```
74
75      # 3) Base velocities (expressed in pelvis frame)
76      v_body = data.sensordata[:, 11:14]  # (B,3) already in body frame
77      w_world = data.qvel[:, 3:6]        # (B,3) world frame
78      w_body = quat_rotate(base_quat_conj, w_world)  # (B,3)
79
80      # 4) Task-specific signals
81      speed_target = torch.full((B, 1), 5.0, device=device, dtype=dtype)
82      speed_error = v_body[:, 0:1] - speed_target  # (B,1)
83
84      # 5) Contacts
85      foot_contact = data.sensordata[:, 0:2]  # (B,2)
86
87      # 6) Feet positions relative to pelvis, in pelvis frame
88      left_foot_world = data.site_xpos[:, 1, :]   # (B,3)
89      right_foot_world = data.site_xpos[:, 2, :]   # (B,3)
90      lf_rel_world = left_foot_world - base_pos
91      rf_rel_world = right_foot_world - base_pos
92      lf_in_base = quat_rotate(base_quat_conj, lf_rel_world)  # (B,3)
93      rf_in_base = quat_rotate(base_quat_conj, rf_rel_world)  # (B,3)
94
95      # 7) Controlled joint subset (positions and velocities)
96      # qpos indices
97      qpos_idx = [
98          8, 9, 10, 11,      # L leg: hip_roll, hip_pitch, knee, ankle
99          13, 14, 15, 16,    # R leg: hip_roll, hip_pitch, knee, ankle
100         17,                # torso
101         18, 21,            # L arm: shoulder_pitch, elbow
102         47, 50,            # R arm: shoulder_pitch, elbow
103     ]
104     joint_pos = data.qpos[:, qpos_idx]  # (B,13)
105
106     # qvel indices
107     qvel_idx = [
108         7, 8, 9, 10,       # L leg: hip_roll, hip_pitch, knee, ankle
109         12, 13, 14, 15,    # R leg: hip_roll, hip_pitch, knee, ankle
110         16,                # torso
111         17, 20,            # L arm: shoulder_pitch, elbow
112         46, 49,            # R arm: shoulder_pitch, elbow
113     ]
114     joint_vel = data.qvel[:, qvel_idx]  # (B,13)
115
116     obs = torch.cat(
117         [
118             pelvis_z,            # (B,1)
119             gravity_in_base,     # (B,3)
120             heading_xy,          # (B,2)
121             v_body,              # (B,3)
122             w_body,              # (B,3)
123             speed_target,        # (B,1)
124             speed_error,         # (B,1)
125             foot_contact,        # (B,2)
126             lf_in_base,          # (B,3)
127             rf_in_base,          # (B,3)
128             joint_pos,           # (B,13)
129             joint_vel,           # (B,13)
130         ],
131         dim=-1,
132     )
133
134     return obs
```

**LOAM-Designed Action Model for `h1hand-run-v0`**

```python
 1   # Action expansion for h1hand-run-v0 (final)
 2   # - Compact action_dim = 13 focused on locomotion and balance:
 3   #    Legs (8): per leg [hip_roll, hip_pitch, knee, ankle] - hip_yaw fixed to 0 for
            straight running.
 4   #    Torso (1): torso actuator for balance.
 5   #    Arms (4): per arm [shoulder_pitch, elbow] for rhythmic swing; shoulder_roll
          via mild synergy; shoulder_yaw/wrist_yaw fixed to 0.
 6   # - Hands (20 actuators per both hands) are set to 0 (neutral) - not needed for
          running.
 7   # - Mild synergy:
 8   #      left_shoulder_roll = -0.2 * left_hip_roll_action
 9   #      right_shoulder_roll = -0.2 * right_hip_roll_action
10   # - Batch/device safe with shape validation.
11   def compute_action(action, data):
12       import torch
13
14       # Normalize action input shape to (B, 13)
15       single = action.dim() == 1
16       if single:
17           action = action.unsqueeze(0)
18
19       B = action.shape[0]
20       action_dim = 13
21       if action.shape[-1] != action_dim:
22           raise ValueError(f"Expected action dim {action_dim}, got {tuple(action.
         shape)}")
23
24       device = action.device
25       dtype = action.dtype
26
27       # Prepare output (B, 61)
28       action_out = torch.zeros((B, 61), device=device, dtype=dtype)
29
30       # Split low-dim actions
31       # Legs
32       l_leg = action[:, 0:4]   # [hip_roll, hip_pitch, knee, ankle]
33       r_leg = action[:, 4:8]   # [hip_roll, hip_pitch, knee, ankle]
34       # Torso
35       torso = action[:, 8:9]   # [torso]
36       # Arms
37       l_arm = action[:, 9:11]  # [shoulder_pitch, elbow]
38       r_arm = action[:, 11:13] # [shoulder_pitch, elbow]
39
40       # Mild balance synergy for shoulder roll from hip roll
41       l_sh_roll = -0.2 * l_leg[:, 0:1]
42       r_sh_roll = -0.2 * r_leg[:, 0:1]
43
44       # Map to 61 actuators
45       # Legs (0-9)
46       action_out[:, 0] = 0.0             # left_hip_yaw
47       action_out[:, 1] = l_leg[:, 0]     # left_hip_roll
48       action_out[:, 2] = l_leg[:, 1]     # left_hip_pitch
49       action_out[:, 3] = l_leg[:, 2]     # left_knee
50       action_out[:, 4] = l_leg[:, 3]     # left_ankle
51       action_out[:, 5] = 0.0             # right_hip_yaw
52       action_out[:, 6] = r_leg[:, 0]     # right_hip_roll
53       action_out[:, 7] = r_leg[:, 1]     # right_hip_pitch
54       action_out[:, 8] = r_leg[:, 2]     # right_knee
55       action_out[:, 9] = r_leg[:, 3]     # right_ankle
56
```

```
57      # Torso (10)
58      action_out[:, 10] = torso[:, 0]      # torso
59
60      # Arms (11-20)
61      action_out[:, 11] = l_arm[:, 0]      # left_shoulder_pitch
62      action_out[:, 12] = l_sh_roll[:, 0] # left_shoulder_roll (synergy)
63      action_out[:, 13] = 0.0              # left_shoulder_yaw
64      action_out[:, 14] = l_arm[:, 1]      # left_elbow
65      action_out[:, 15] = 0.0              # left_wrist_yaw
66      action_out[:, 16] = r_arm[:, 0]      # right_shoulder_pitch
67      action_out[:, 17] = r_sh_roll[:, 0] # right_shoulder_roll (synergy)
68      action_out[:, 18] = 0.0              # right_shoulder_yaw
69      action_out[:, 19] = r_arm[:, 1]      # right_elbow
70      action_out[:, 20] = 0.0              # right_wrist_yaw
71
72      # Hands (21-60) remain zeros (neutral/passive)
73
74      # Validate final shape
75      if action_out.shape != (B, 61):
76          raise RuntimeError(f"Expanded action has wrong shape {tuple(action_out.
        shape)}, expected {(B, 61)}")
77
78      # Restore original shape if single sample input
79      if single:
80          action_out = action_out.squeeze(0)
81
82      return action_out
```

### B.2.3. ANALYSIS OF LOAM-DESIGNED OBSERVATION MODEL (48 DIMENSIONS)

The LOAM-Designed model redesigns the observation space to be more compact and semantically meaningful. The function reduces the observation space from 151 to 48 dimensions by selecting only the joint states relevant to locomotion—legs, torso, and primary arm joints (lines 96-114). Proprioceptive states from the hands, which are passive during running, are excluded.

The observation is also designed to be invariant to the robot's heading, a modification that aids in learning generalizable locomotion policies. This is achieved by representing key vectors in the pelvis's local coordinate frame. For example, the world gravity vector (lines 66-67), base angular velocities (lines 77-78), and relative foot positions (lines 90-93) are all transformed into this local frame using quaternion rotations.

Furthermore, the function engineers features that directly relate to the task goal. It includes the target speed (5.0 m/s) and the current forward speed error as direct inputs to the policy (lines 68-69). This provides a low-latency signal correlated with the reward function. For stability, the design includes the positions of the feet relative to the pelvis (lines 88-93) and binary foot contact sensor data (line 85), which inform the policy about the robot's base of support and the current phase of the gait cycle.

### B.2.4. ANALYSIS OF LOAM-DESIGNED ACTION MODEL (13 → 61 DIMENSIONS)

Complementing the observation space, the LLM structured the action space around a low-dimensional latent representation. The policy learns to control a 13-dimensional vector that is deterministically expanded to the full 61-dimensional actuator space. This latent vector corresponds to the key joints for locomotion: 8 for the legs, 1 for the torso, and 4 for the primary arm swing (lines 30-38).

The design also embeds a biomechanical prior by hard-coding a contralateral coordination pattern. It couples the shoulder roll action to the opposite hip's roll action (lines 42-43), which is then mapped to the shoulder roll actuators (lines 41-42, 62 and 67). This design generates a stabilizing arm swing that counter-rotates the torso against leg movements, removing the need for the agent to learn this fundamental coordination from scratch. Finally, actuators for joints not directly involved in running, such as those in the hands or for hip/shoulder yaw, are fixed to zero (lines 28, 46, 51, 63, 65, 68, and 70). This

focuses the agent's learning on the essential degrees of freedom for the gait.

## B.3. Case Study 2: `Isaac-Lift-Cube-Franka-v0` (Expand Action Dimension)

The objective of this task is to command the Franka robotic arm to lift a cube from the table and transport it to a target position. This task primarily evaluates precise end-effector coordination, grasp control, and multi-joint manipulation behavior.

### B.3.1. LOAM-DESIGNED ACTION MODEL CODE FOR ISAAC-LIFT-CUBE-FRANKA-V0

---

**LOAM-Designed Action Model for `Isaac-Lift-Cube-Franka-v0`**

```python
 1  # Action mapping summary
 2  # ---------------------
 3  # Chosen action_dim: 12
 4  # Rationale:
 5  # - 8 direct channels preserve full actuator authority (7 arm joints + 1 gripper)
       .
 6  # - +3 latent "synergy" scalars (shoulder, elbow, wrist) create smooth,
       correlated sub-motions across joint groups, often improving learnability in
       manipulation.
 7  # - +1 nullspace/posture scalar drives the arm toward a comfortable nominal
       configuration without constraining dexterity.
 8  # Mapping approach:
 9  # - a[0:7]: base joint commands (normalized, passed through tanh for safety).
10  # - a[7]:   base gripper command (normalized).
11  # - a[8:11]: synergy scalars applied via fixed synergy vectors over joints (
       gripper unaffected).
12  # - a[11]:  posture scalar scales a term toward a nominal joint posture using
       current joint_pos from data.
13  # Implementation details:
14  # - Batch-safe: accepts (B,12) or (12,), returns (B,8) or (8,).
15  # - Device-safe: all new tensors created on the same device/dtype as input `
       action`.
16  # - Validations for shape consistency and safe clamping to [-1, 1].
17  def compute_action(action, data):
18      import torch
19
20      if not torch.is_tensor(action):
21          raise TypeError("`action` must be a torch.Tensor")
22
23      device = action.device
24      dtype = action.dtype
25
26      # Accept 1D or 2D action; normalize to (B, D)
27      squeezed = False
28      if action.dim() == 1:
29          action = action.unsqueeze(0)
30          squeezed = True
31      if action.dim() != 2:
32          raise ValueError(f"`action` must be 1D or 2D, got shape {tuple(action.
       shape)}")
33      B, D = action.shape
34      expected_dim = 12
35      if D != expected_dim:
36          raise ValueError(f"Expected action_dim={expected_dim}, got {D}")
37
38      # Extract required state for posture term
39      if 'joint_pos' not in data:
40          raise KeyError("data must contain 'joint_pos' for posture shaping")
41      qpos = data['joint_pos']
```

---

```
42      if qpos.dim() != 2 or qpos.shape[1] < 7:
43          raise ValueError(f"data['joint_pos']␣must␣be␣of␣shape␣(B,␣>=7),␣got␣{
        tuple(qpos.shape)}")
44      if qpos.shape[0] != B:
45          raise ValueError(f"Batch␣mismatch:␣action␣batch␣{B}␣vs␣data['joint_pos']␣
        batch␣{qpos.shape[0]}")
46
47      qpos_arm = qpos[:, :7].to(device=device, dtype=dtype)  # (B,7)
48
49      # Base actuator commands
50      base_joint = torch.tanh(action[:, 0:7])    # (B,7)
51      base_grip = torch.tanh(action[:, 7:8])     # (B,1)
52
53      # Latent synergy scalars
54      s_shoulder = torch.tanh(action[:, 8:9])    # (B,1)
55      s_elbow    = torch.tanh(action[:, 9:10])   # (B,1)
56      s_wrist    = torch.tanh(action[:, 10:11])  # (B,1)
57      s_posture  = torch.tanh(action[:, 11:12])  # (B,1)
58
59      # Fixed synergy vectors over actuators (length 8, last element zero to leave
        gripper unaffected)
60      syn_shoulder = torch.tensor([0.80, 0.60, -0.20, 0.00, 0.00, 0.00, 0.00,
        0.00],
61                                   device=device, dtype=dtype).view(1, 8)
62      syn_elbow    = torch.tensor([0.00, 0.00, 0.60, 0.90, 0.20, 0.00, 0.00, 0.00],
63                                   device=device, dtype=dtype).view(1, 8)
64      syn_wrist    = torch.tensor([0.00, 0.00, 0.00, 0.00, 0.50, 0.60, 0.60, 0.00],
65                                   device=device, dtype=dtype).view(1, 8)
66
67      synergy_gain = torch.tensor(0.35, device=device, dtype=dtype)
68
69      # Combined synergy effect (B,8)
70      synergy_effect = (s_shoulder * syn_shoulder +
71                        s_elbow    * syn_elbow +
72                        s_wrist    * syn_wrist) * synergy_gain
73
74      # Nullspace/posture term toward a comfortable nominal Franka posture
75      q_nom = torch.tensor([0.0, -0.7853982, 0.0, -2.3561945, 0.0, 1.5707963,
        0.7853982],
76                            device=device, dtype=dtype).view(1, 7).expand(B, -1)
77      posture_gain = torch.tensor(0.25, device=device, dtype=dtype)
78      posture_term = posture_gain * s_posture * torch.tanh(q_nom - qpos_arm)  # (B
        ,7)
79
80      # Compose final actuator commands
81      joint_out = base_joint + synergy_effect[:, :7] + posture_term  # (B,7)
82      gripper_out = base_grip + synergy_effect[:, 7:8]               # (B,1);
        synergy part is zero
83
84      action_out = torch.cat([joint_out, gripper_out], dim=1)  # (B,8)
85      action_out = torch.clamp(action_out, -1.0, 1.0)
86
87      # Final shape validations
88      if action_out.shape != (B, 8):
89          raise RuntimeError(f"Internal␣mapping␣error␣produced␣shape␣{tuple(
        action_out.shape)},␣expected␣{(B,␣8)}")
90      if squeezed:
91          action_out = action_out.squeeze(0)  # return (8,) if input was (12,)
92      return action_out
```

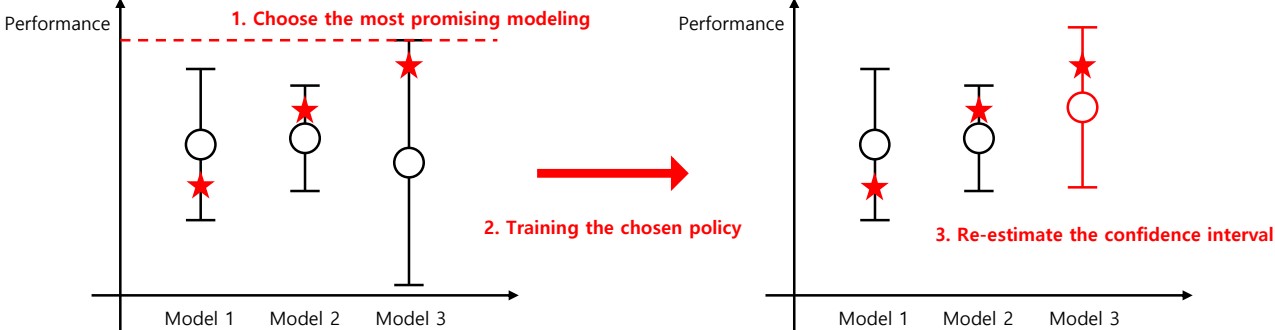

*Figure 14.* Illustration of the OFU-based selection process. At each iteration, (1) the model with the highest upper confidence bound is chosen, (2) its policy is further trained for a fixed step, and (3) the confidence interval is re-estimated. This optimistic rule ensures a balance between exploiting promising models and continuing to explore those that may turn out to be optimal.

### B.3.2. ANALYSIS OF LOAM-DESIGNED ACTION MODEL (12 → 8 DIMENSIONS)

Unlike the locomotion tasks where LOAM often compresses the action space by disabling unnecessary joints, the generated action model for `Isaac-Lift-Cube-Franka-v0` instead expands the original 8-dimensional action space into a 12-dimensional latent action interface. This demonstrates that LOAM does not enforce a fixed low-dimensional prior, but adapts the action structure depending on task requirements.

The generated model preserves the original 8 actuator commands of the Franka arm while introducing four additional latent control channels. Specifically, three latent dimensions are designed as synergy controls for the shoulder, elbow, and wrist joint groups, enabling coordinated sub-motions across related actuators. Rather than independently controlling each joint, these synergy channels encourage smooth and correlated manipulation behaviors that are often beneficial for robotic manipulation tasks.

In addition, the generated model introduces a posture-related latent dimension that biases the arm toward a nominal comfortable configuration using the current joint positions. Importantly, this posture term does not directly constrain the reachable workspace or dexterity of the robot, but instead provides a soft null-space shaping prior that can stabilize arm configurations during manipulation.

Overall, the generated action model illustrates that LOAM can restructure action interfaces beyond simple masking or dimensionality reduction. Depending on the task semantics and morphology, the LLM may preserve, compress, or expand the action space to construct more structured and learnable control interfaces.

## C. Exploration-Exploitation Based on Optimism

To construct LOAM-Race, we adopt optimism-in-the-face-of-uncertainty (OFU), a widely used strategy for balancing exploration and exploitation. This OFU-based rule efficiently narrows the set of promising candidates while maintaining a natural trade-off between exploration and exploitation, as illustrated in Figure 14.

At each iteration, we estimate the upper confidence bound of every candidate model from its estimated performance and associated uncertainty, and then select the model with the highest bound. The chosen policy is then trained for a fixed step, which refines its performance estimate and reduces uncertainty. This selection rule inherently balances exploration—by occasionally sampling uncertain candidates—and exploitation—by favoring those with strong empirical performance.

## D. Raw observation

### D.1. HumanoidBench

In our HumanoidBench experiments, the raw observation space, $\Omega_{\text{raw}}$, is provided to the LLM consists of physical states accessible through the `mj.Data` interface as follows:

- `qpos`: Generalized positions (joint angles, base pose).

- `qvel`: Generalized velocities (joint rates, base velocity).

- `site_xpos`: 3D position of each defined site (useful for end-effectors, sensors, etc.).

- `site_xmat`: 3x3 rotation matrix representing the orientation of each site (flattened).

- `qfrc_actuactor`: Actuator forces mapped to degrees of freedom.

- `xpos`: 3D position of each body.

- `xquat`: 4D orientation (quaternion) of each body.

- `xmat`: 3x3 rotation matrix representing the orientation of each body (flattened).

- `cvel`: 6D spatial velocity (linear & angular) of each body.

- `cfrc_ext`: 6D external wrench (force & torque) on each body.

- `cinert`: 10D composite rigid body inertia of each body.

- `actuator_force`: Force/torque generated by each actuator.

- `sensordata`: Scalar sensor outputs.

## D.2. Isaac Lab

In our Isaac Lab experiments, the raw observation space, $\Omega_{\text{raw}}$, is provided to the LLM consists of physical states accessible through the `isaaclab.assets.RigidObjectData` interface.

- `joint_pos`: Joint positions of all joints.

- `joint_vel`: Joint velocities of all joints.

- `joint_acc`: Joint acceleration of all joints.

- `root_link_pos_w`: Root link position in world frame.

- `root_link_quat_w`: Root link orientation $(w, x, y, z)$ in world frame.

- `root_link_lin_vel_w`: Root link linear velocity in world frame.

- `root_link_ang_vel_w`: Root link angular velocity in world frame.

- `root_link_lin_vel_b`: Root link linear velocity in base frame.

- `root_link_ang_vel_b`: Root link angular velocity in base frame.

- `root_link_state_w`: Root link state $[\text{pos}, \text{quat}, \text{lin\_vel}, \text{ang\_vel}]$ in world frame.

- `root_state_w`: Alias for root link state in world frame.

- `root_com_pos_w`: Root center of mass position in world frame.

- `root_com_quat_w`: Root center of mass orientation $(w, x, y, z)$ in world frame.

- `root_com_lin_vel_w`: Root center of mass linear velocity in world frame.

- `root_com_ang_vel_w`: Root center of mass angular velocity in world frame.

- `root_com_lin_vel_b`: Root center of mass linear velocity in base frame.

- `root_com_ang_vel_b`: Root center of mass angular velocity in base frame.

- `root_com_vel_w`: Root center of mass velocity $[\text{lin\_vel}, \text{ang\_vel}]$ in world frame.

- `root_com_state_w`: Root center of mass state [pos, quat, lin_vel, ang_vel] in world frame.

- `body_link_pos_w`: Positions of all bodies' link frames in world frame.

- `body_link_quat_w`: Orientation $(w, x, y, z)$ of all bodies' link frames in world frame.

- `body_link_lin_vel_w`: Linear velocity of all bodies' link frames in world frame.

- `body_link_ang_vel_w`: Angular velocity of all bodies' link frames in world frame.

- `body_link_vel_w`: Body link velocity [lin_vel, ang_vel] in world frame.

- `body_link_state_w`: State of all bodies' link frame [pos, quat, lin_vel, ang_vel] in world frame.

- `body_state_w`: Alias for body link state in world frame.

- `body_com_pos_w`: Positions of all bodies' center of mass in world frame.

- `body_com_quat_w`: Orientation $(w, x, y, z)$ of the principle axis of inertia of all bodies in world frame.

- `body_com_lin_vel_w`: Linear velocity of all bodies' center of mass in world frame.

- `body_com_ang_vel_w`: Angular velocity of all bodies' center of mass in world frame.

- `body_com_vel_w`: Body center of mass velocity [lin_vel, ang_vel] in world frame.

- `body_com_state_w`: State of all bodies' center of mass [pos, quat, lin_vel, ang_vel] in world frame.

- `projected_gravity_b`: Projection of the gravity direction on the base frame.

- `heading_w`: Yaw heading of the base frame (in radians).

- `generated_commands`: The desired goal pose for the target object (e.g., cube) specified in the environment (world) frame.

## E. More Details of NVIDIA Isaac Lab Experiments

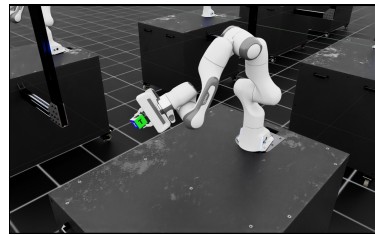 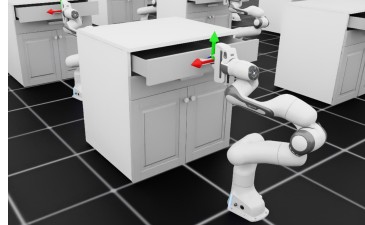 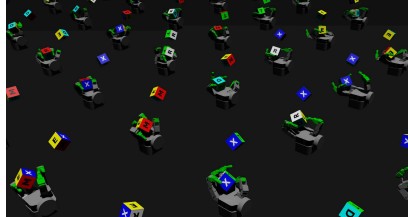

*(a)* Isaac-Lift-Cube-Franka-v0    *(b)* Isaac-Open-Drawer-Franka-v0    *(c)* Isaac-Repose-Cube-Allegro-v0

*Figure 15.* Examples of Isaac Lab tasks.

Beyond the MuJoCo and Humanoid morphologies in HumanoidBench, we extend our evaluation to NVIDIA Isaac Lab (Mittal et al., 2023). This environment offers a distinct dynamics and morphologies, such as Franka Emika Panda (7-DoF single-arm manipulator) and the Allegro Hand (16-DoF dexterous hand).

We evaluated LOAM on the following tasks (Figure 15):

- **Isaac-Lift-Cube-Franka-v0**: Pick a cube and bring it to a sampled target position with the Franka robot. (handcrafted observation: 36 dimension, raw observation: 1,152 dimension)

- **Isaac-Open-Drawer-Franka-v0**: Grasp the handle of a cabinet's drawer and open it with the Franka robot. (handcrafted observation: 31 dimension, raw observation: 1,801 dimension)

- **Isaac-Repose-Cube-Allegro-v0**: In-hand reorientation of a cube using Allegro hand. (handcrafted observation: 72 dimension, raw observation: 1,928 dimension)

In our Isaac Lab experiments, we define the raw observation space ($\Omega_{\text{raw}}$) to consist solely of fundamental physical states accessible directly through the `isaaclab.assets.RigidObjectData` API as in HumanoidBench, while FastTD3 uses a handcrafted observation provided by Isaac Lab. We include **every** dynamic state variable available in the API—excluding only static constants and mathematically identical duplicates—thereby exposing the LLM to the complete, uncurated physical state of the system. A comprehensive list of these features is provided in Appendix D.2.

Note that we intentionally exclude high-level, pre-processed features—typically provided in default Isaac Lab environments—from the raw observation space. For instance, the default (handcrafted) observation space for the Allegro Hand task includes computed values such as `goal_quat_diff`—a derived feature that explicitly calculates the quaternion difference between the current and target orientations to simplify the learning problem.

In contrast, LOAM is restricted to raw absolute data, such as the root link's orientation in the world frame (`root_link_quat_w`) and the target command (`generated_commands`). Consequently, the LOAM-generated observation function is required to discover and implement the necessary mathematical transformations (e.g., quaternion multiplication for relative orientation) to extract task-relevant information, rather than relying on human-engineered shortcuts.

## F. Ablation Studies

### F.1. Ablation Study in Observation and Action Models

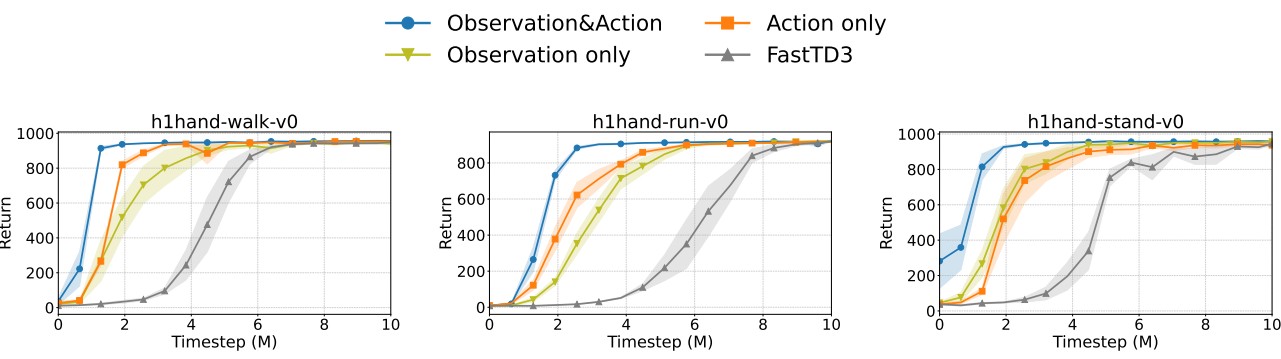

*Figure 16.* Comparison in the effect of observation and action models on HumanoidBench `stand`, `walk`, and `run` tasks.

We compare the full LOAM framework against observation-only and action- only variants on locomotion tasks. Note that the action-only variant, like the FastTD3 baseline, uses handcrafted observations instead of the high-dimensional raw ones. Each component improves over the baseline individually, but jointly optimizing both observation and action spaces in the full LOAM framework yields the fastest convergence and largest gain.

### F.2. Comparison in Different Candidate Selection Strategies of LOAM-Race

To analyze the impact of both the acquisition function and the resource allocation strategy in LOAM-Race, we compare our default Bayesian Ridge Regression (BRR)-based racing approach against several alternatives.

**LOAM-Race (Bayesian Ridge)** is our default method, which uses Bayesian Ridge Regression to extrapolate future return trajectories and estimate optimistic upper confidence bounds for candidate selection. **LOAM-Race (Gaussian Process)** replaces BRR with Gaussian Process Regression (GPR) using an RBF kernel, which can capture non-linear trajectory patterns. We utilized Scikit-learn (Pedregosa et al., 2011) to implement both acquisition functions. We additionally compare against two Successive Halving (Jamieson & Talwalkar, 2016)-based baselines: **LOAM-SHA4** and **LOAM-SHA8**, which are initialized with 4 and 8 candidate models, respectively. These methods periodically eliminate the lower-performing half of the candidates and reallocate the remaining training budget to the surviving candidates. Finally, we evaluate **LOAM-CBS**

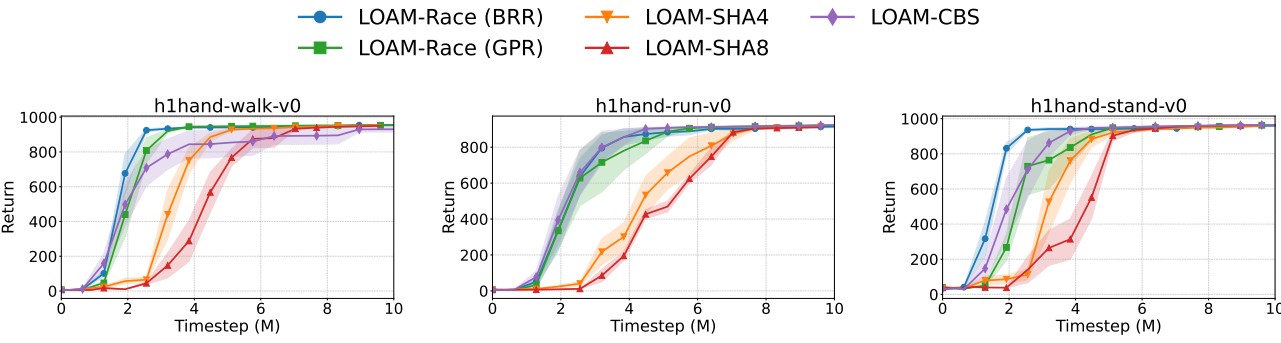

*Figure 17.* Learning curves of LOAM-Race with different candidate selection strategies on HumanoidBench `stand`, `walk`, and `run` tasks.

(**Current Best Selection**), a greedy baseline that always allocates the next training budget to the candidate with the highest current evaluation return.

We evaluated those variants on the basic locomotion tasks (`stand`, `walk`, `run`) in HumanoidBench. As shown in 17, BRR achieves the fastest and most stable convergence among the acquisition functions considered. Although GPR provides a more expressive non-linear estimator, it offers little practical advantage over BRR despite its increased modeling complexity. Among different resource allocation strategies, the racing-based approaches consistently achieve the best overall performance. Specifially, the SHA-based methods converge more slowly in our setting, suggesting that their fixed elimination schedule is less suitable for continuous candidate evaluation during RL training. Overall, these results indicate that LOAM-Race effectively balances exploration and exploitation, enabling robust and sample-efficient candidate selection

Animated examples of both the CBS and LOAM-Race selection processes, including dynamic competition between candidates and cases where initially underperforming candidates are eventually selected, are available on the project page: https://sites.google.com/view/icml2026-loam.

### F.3. Comparison in LLM Backbone

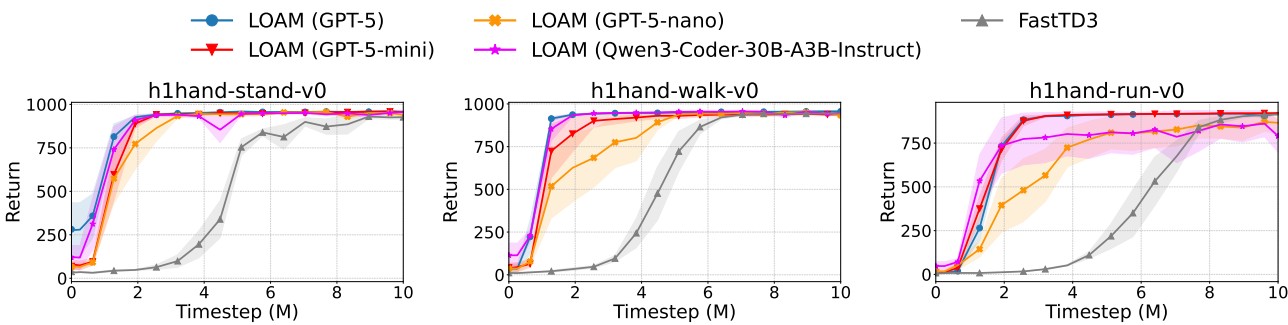

*Figure 18.* Learning curves of LOAM with different LLM backbones on HumanoidBench `stand`, `walk`, and `run` tasks.

To address concerns regarding the dependency of our framework on state-of-the-art proprietary models (e.g., GPT-5), we conducted an ablation study using varying LLM sizes and an open-weights model. We compared the standard GPT-5 [5] against lighter variants (GPT-5-mini [6], GPT-5-nano [7]) and the open-weights Qwen3-Coder-30B-A3B-Instruct (Yang et al.,

---

[5]The flagship model with the highest parameter count and reasoning capability. We use this as the upper-bound baseline to evaluate the maximum potential of our framework.

[6]A mid-sized model optimized for a balance between cost and performance. It retains significant reasoning abilities while offering lower latency compared to the GPT-5.

[7]The most lightweight variant, designed for extreme efficiency and low-latency applications. While it has the lowest reasoning capacity among the three, it serves to test the feasibility of our method in resource-constrained or on-device settings.

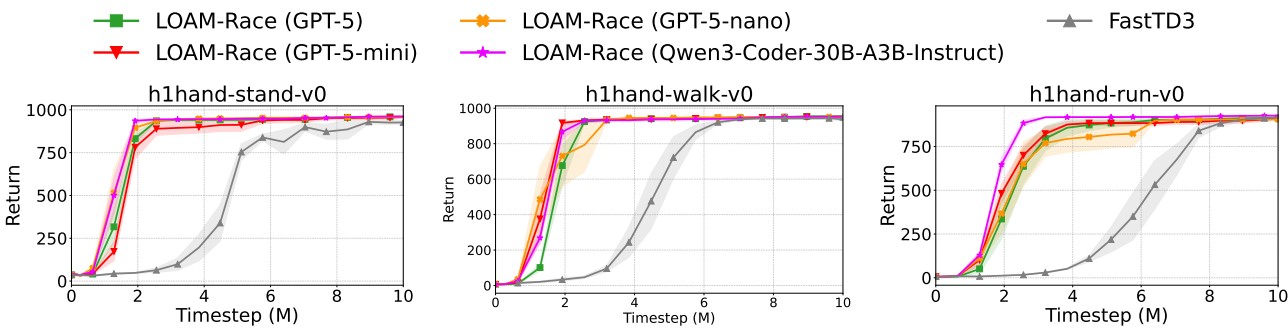

*Figure 19.* Learning curves of LOAM-Race with different LLM backbones on HumanoidBench `stand`, `walk`, and `run` tasks.

2025), analyzing their performance relative to the FastTD3 baseline.

As illustrated in Figures 18 and 19, a consistent pattern emerges across all tasks: regardless of the LLM backend employed—from the large-scale GPT-5 to the compact GPT-5-nano—both LOAM and LOAM-Race substantially outperform the non-LLM baseline (FastTD3). This demonstrates that semantic guidance from LLMs, even those with constrained capacity, provides a fundamental advantage in accelerating policy learning compared to approaches relying exclusively on handcrafted observation-action models.

However, while performance remains robust on simpler tasks (e.g., `stand` and `walk`), we observe a notable divergence in the more challenging `run` task under the standard LOAM framework (Figure 18). Specifically, models with reduced reasoning capabilities—such as GPT-5-nano and Qwen3-Coder-30B—exhibit marked performance degradation and delayed convergence compared to GPT-5. The gap widens significantly after approximately 4M timesteps, with GPT-5-nano and Qwen3-Coder plateauing at lower asymptotic returns.

Critically, the LOAM-Race framework (Figure 19) not only addresses this sensitivity to LLM capability but, remarkably, enables smaller models to match or even exceed the performance of their larger counterparts. In the `run` task, LOAM-Race successfully narrows the performance gap, with the open-weights Qwen3-Coder-30B achieving returns comparable to—and in some phases surpassing—GPT-5. More strikingly, in the `stand` task, Qwen3-Coder-30B consistently outperforms all proprietary models throughout training, demonstrating that competitive racing dynamics can effectively leverage the capabilities of open-source LLMs. These results underscore a key contribution of the LOAM-Race mechanism: it enhances framework robustness by reducing dependence on high-capacity LLMs while simultaneously unlocking the full potential of smaller models through collaborative refinement, thereby demonstrating the practical viability of deploying our method with more accessible, cost-effective, and open-source language models.

### F.4. Racing vs. Iterative Refinement

To validate the design choice of our candidate selection mechanism, we compare our proposed method, **LOAM-Race**, against an alternative iterative strategy named **LOAM-Refine**. Unlike LOAM-Race, which evaluates candidates in parallel, LOAM-Refine attempts to iteratively improve the observation and action space definitions during training.

A refinement step is triggered if the agent's return fails to exceed the current maximum for 5 consecutive checkpoints. Modifying the observation or action specifications inevitably alters the input or output dimensions (or semantics) of the underlying policy and value networks. Since transferring learned weights between neural networks with mismatched architectures is non-trivial, we re-initialize the agent's parameters and restart training from scratch after each refinement.

The experimental results in Figure 20 highlight the inherent limitations of this iterative approach. As shown in Seed 0 (Figure 20a) and Seed 2 (Figure 20c), the mandatory resets cause a fragmentation of the total training budget. While LOAM-Race allows a candidate to utilize the full 50M timesteps to reach asymptotic convergence, LOAM-Refine splits this budget into shorter segments. The agent is repeatedly interrupted just as it begins to learn, losing all prior experience and being forced to relearn basic locomotion skills from scratch. This inefficiency prevents the policy from mastering complex behaviors within the limited time remaining after a reset.

Furthermore, the iterative strategy proves ineffective when the initial candidate is fundamentally flawed. In Seed 1

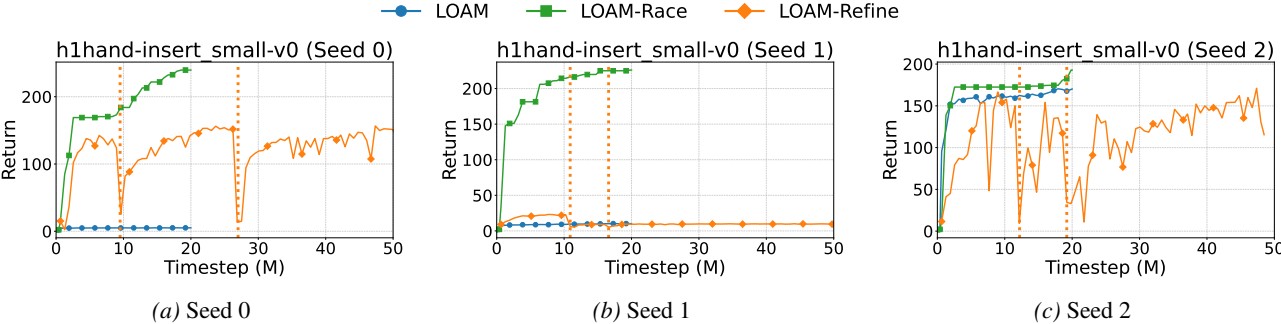

*Figure 20.* Performance comparison between LOAM-Race (Ours) and LOAM-Refine. The green line represents the average performance of LOAM-Race, while the orange line shows individual runs of LOAM-Refine across three random seeds. Orange vertical dotted lines indicate the timesteps where the observation-action space was refined.

(Figure 20b), where the starting design yields near-zero returns, subsequent refinements fail to recover a viable solution. The combination of a poor starting point and the sample inefficiency of repeated resets results in a complete failure, confirming that iteratively searching for a design is less robust than LOAM-Race's parallel selection. By generating candidates upfront and evaluating them without structural interruptions, LOAM-Race ensures that the best candidate benefits from the maximum available sample budget, avoiding the wasteful cold starts associated with iterative refinement.

### F.5. Additional guidance in prompt

We append additional guidance to prompts used for LLM-based observation and action modeling. First, we modify the task descriptions in Humanoid-Bench into more precise definitions. For example, in 'h1hand-reach-v0', the description *"Reach a randomly initialized 3D point with the left hand."* can be misread as involving only the hand, whereas the task requires whole-body movement toward a distant target. Without clarification, an LLM may ignore the rest of the body in observation and action modeling. To prevent this, we restate the task as *"Touch a target point in space using its left hand. It must accomplish this while maintaining a stable and upright posture."* Second, we add a line to reasoning guideline, emphasizing that whole-body posture is an implicit requirement for any physical task; *"For any physical task, maintaining the robot's stable and balanced posture is a fundamental and implicit requirement. The final code must reflect a deep understanding of*

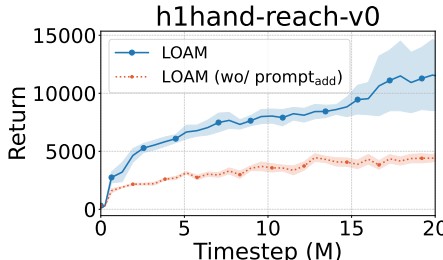

*Figure 21.* Ablation on the additional guidance in prompts.

*the need for whole-body postural control while performing the primary task."*. We observed that these changes ensure the LLM interprets and models the task correctly, capturing both the target objective and the necessary full-body constraints, resulting in higher performance across some tasks such as `reach` (Figure 21).

