# OpenReview forum: "Designing Observation and Action Models for Efficient Reinforcement Learning with LLMs"
_ICML.cc/2026/Conference — ICML 2026 regular_

### Official Review · Reviewer_11eZ · 2026-03-11

**Soundness:** 3
**Presentation:** 3
**Significance:** 2
**Originality:** 2
**Overall Recommendation:** 4
**Confidence:** 4

**Summary:**

This paper proposes LOAM, a framework that uses an LLM to generate observation and action wrappers mapping raw environment spaces into agent-facing spaces. The authors also propose LOAM-Race, which allocates training budget across multiple LLM-generated candidates using optimistic return extrapolation. Experiments on HumanoidBench and Isaac Lab show improvements in sample efficiency / final performance over FastTD3 and, on HumanoidBench, over LESR as well.

**Compliance With Llm Reviewing Policy:**

Affirmed.

**Final Justification:**

I would like to thank the author for a convincing rebuttal. I have my score to a 4. The conceptual novelty, particularly for action construction, remains somewhat limited, but the LOAM-Race mechanism is a meaningful enough contribution that I think the bar for novelty has been borderline reached. The additional experiments solve any soundness issues.

**Key Questions For Authors:**

1. How much task-specific human effort is required to prepare the prompt inputs for a new domain beyond a raw simulator API?
2. How much of the performance gain remains if the prompts do not include detailed actuator maps, morphology descriptions, and explicit posture/balance instructions?
3. Why is Isaac Lab only compared against FastTD3 rather than also against LESR or other relevant baselines?
4. Why are there no comparisons against simpler racing/allocation baselines such as current-best selection or successive halving?
5. How robust is LOAM-Race to settings where a simple greedy allocation strategy would fail due to misleading early returns?
6. Would LOAM-Race still identify the best candidate when strong designs improve only late in training, i.e. when greedy early selection would fail?

**Limitations:**

yes

**Strengths And Weaknesses:**

## Strengths

- The paper is in a yet underappreciated field. Most LLM-for-RL work has focused on reward design, and observation/action design is a natural extension.
- The empirical results are strong.
- The LOAM racing formulation is easy to understand and likely reusable.
- The paper includes useful ablations, including observation-only vs observation+action, number of candidates, acquisition function, and LLM backbone.

## Weaknesses

- The novelty is overstated. At a high level, the method prompts an LLM to generate wrapper code, instantiates candidates, and allocates more training to promising ones. This is reasonable systems work but not a major new RL idea.
- The prompts encode substantial expert structure: task descriptions, robot morphology, detailed variable/index mappings, actuator maps, explicit instructions about posture/balance, and guidance about choosing action dimensions. Prompt engineering seems to be doing a lot of the work.
- As a follow up from the last weakness, the paper does not discuss the action masking literature [1]. Including a baseline which performs action masking would also strengthen the baselines.
- There is no ablation on action only. I suspect the contribution of action wrappers (which essencially distills to action masking) is heavily contributing to the seen performance. The authors should justify why they do not include this baseline.
- The paper frames LOAM as designing from raw observation/action spaces, but the raw spaces are defined in a way that favors the proposed approach, and the appendix shows pretty heavy prompt engineering. The baselines are also quite weak. On Isaac Lab, the comparison is only against FastTD3.
- The statistical significance of the results is somewhat suspect. Reporting 3 seeds is weak given the claimed stochasticity of LLM-generated designs.


[1] Huang, Shengyi, and Santiago Ontañón. "A closer look at invalid action masking in policy gradient algorithms." arXiv preprint arXiv:2006.14171 (2020).

---

> ### Author Rebuttal · Authors · 2026-03-31
>
> We thank for the thorough review. We have conducted additional experiments. (https://sites.google.com/view/icml-loam.)
>
> ### **[W1] Novelty and contribution**
>
> We agree that LOAM does not propose a new RL algorithm in the conventional sense and will calibrate this claim. However, we do not believe this diminishes its novelty.
>
> Our contribution lies in the emerging LLM+RL direction. While prior work has focused mainly on reward design, LOAM studies observation/action modeling, a relatively underexplored direction for improving RL on new tasks. We do not claim a new optimization principle, but novelty in how LLMs can improve RL training.
>
> Concretely, LOAM does not introduce a new learner, but makes observation/action modeling practical, enabling more efficient learning on new tasks. We view this as a meaningful contribution: automating interface design and identifying strong designs within a standard training budget.
>
> ### **[W2,Q1] Prompt structure, and human effort**
>
> LOAM does **not** rely on task-specific human expert knowledge, but it does assume semantically meaningful environment information. This scope will be stated more explicitly in the paper.
>
> The prompt contains: (1) **generic guidance** that we design and is reusable across different tasks and environments, and (2) **environment-derived semantics** such as variable descriptions, morphology, and actuator information from simulator documentation and XML files. The first part guides the LLM to reason carefully, but does not tell it which variables/actions to choose. The second part is not extra expert knowledge; it is the same information a human designer would also need.
>
> The required human effort is minimal, as only the **environment-derived semantics** need to be adapted when the setting changes. If only the task changes, the user writes a short task description (objective, initialization, termination). If the simulator or robot changes, the user collects the relevant semantic information from documentation/XML and places it into our prompt format. This step is straightforward; please see jbXo [W1].
>
> ### **[Q2] Weakly semantic settings**
>
> Please see jmEC [W1,Q1]).
>
> ### **[W3,W4] Relation to action masking and action-only contribution**
>
> We agree that this point should be clarified better. LOAM's action modeling is different from standard action masking. Please see  jmEC [W3,Q2].
>
> During the rebuttal, we add an action-only ablation on Basic Locomotion tasks (stand, walk, run) with 5 seeds in the link (2. Ablation on action only). LOAM (obs-only) and LOAM (act-only) achieve similar performance, and the strongest performance is obtained when both are used together. This suggests that the gains cannot be attributed mainly to action wrappers alone.
>
> ### **[W5,Q3] Isaac Lab baseline comparison**
>
> During the rebuttal, we conducted additional experiments with LESR on Isaac Lab in the link (3. LESR on Isaac Lab) and observe that LOAM-Race still outperforms LESR, although the performance gap is smaller than in HumanoidBench.
>
> We attribute this to the relatively simpler nature of Isaac Lab tasks, which involve lower-dimensional observation/action spaces (e.g., single-arm or finger-based robots). Moreover, the default observation space (Ω_hand) is already well-engineered. LESR builds upon this by adding extra features, whereas LOAM-Race does not utilize Ω_hand at all. Despite this, LOAM-Race achieves strong performance, suggesting that the LLM can construct effective observation/action representations.
>
> ### **[W6] Statistical significance**
>
> We report 3 seeds in the main paper for consistency with prior baselines (e.g., FastTD3), and include standard error in all plots. To address the reviewer's concern, all rebuttal experiments use 5 seeds and show consistent trends.
>
> ### **[Q4] Comparison with simpler racing baselines**
>
> During rebuttal, we added experiments with current-best selection and SHA in the link (1. Comparison). LOAM-Race (w/ BRR) achieved the best performance among the three methods.
>
> ### **[Q5,Q6] LOAM-Race robustness to misleading early returns**
>
> LOAM-Race uses UCB-based selection (`mu+sigma`) to balance exploration and exploitation. This is precisely intended to avoid the failure mode of greedy allocation under misleading early returns. Candidates with high uncertainty can continue receiving budget even if their early returns are not the best, which helps recover late-improving designs. Our additional experiments confirm that LOAM-Race outperforms current-best selection in this setting.
>
> For further illustration, we provide visual examples of the racing process including comparisons with current-best selection, cases where initially low-performing candidates are eventually selected, and dynamic competition between candidates in the link(4. Visualization). These examples demonstrate how LOAM-Race effectively balances exploration and exploitation over time.

---

> > ### Author Rebuttal · Reviewer_11eZ · 2026-03-31
> >
> > I would like to thank the author for a convincing rebuttal. I will raise my score to a 4. The conceptual novelty, particularly for action construction, remains somewhat limited, but the LOAM-Race mechanism is a meaningful enough contribution that I think the bar for novelty has been borderline reached. The additional experiments solve any soundness issues.
> >
> > I would suggest that for future work, the authors investigate parallel scaling in addition to the sequential switching shown in this work.

---

> > > ### Author Response · Authors · 2026-04-01
> > >
> > > Thank you for the thoughtful feedback and for taking the time to reconsider our submission. We appreciate your positive assessment and are glad that our clarifications and additional experiments addressed your concerns.
> > >
> > > We also thank you for the suggestion regarding parallel scaling—this is an interesting direction, and we will consider it for future work.

---

### Official Review · Reviewer_jmEC · 2026-03-13

**Soundness:** 3
**Presentation:** 2
**Significance:** 3
**Originality:** 2
**Overall Recommendation:** 4
**Confidence:** 4

**Summary:**

This paper proposes the LOAM framework, leveraging the semantic reasoning capabilities of large language models to design refined agent observation and action spaces from high-dimensional raw environments. To mitigate the substantial model selection overhead stemming from the stochasticity and variable quality of LLM-generated outputs, the authors introduce the LOAM-Race mechanism. This mechanism dynamically allocates training resources based on predictive potential scores, thereby identifying strong candidate configurations within a computational budget equivalent to a single model training run.

**Compliance With Llm Reviewing Policy:**

Affirmed.

**Final Justification:**

The authors have adequately clarified that the fixed action mapping does not bottleneck late-stage performance in the fixed-task setting. I raise my score accordingly, though I encourage the authors to explicitly discuss the scope limitation regarding broader task settings in the final version.

**Key Questions For Authors:**

1. The current conclusions rely heavily on structured semantic inputs, lacking evaluations in anonymized-variable or weakly semantic settings. This is crucial for assessing the broad applicability of the proposed method. Could the authors discuss how this affects the claimed breadth of applicability, and clarify this limitation more explicitly?
2. Could the fixed action mapping impose an upper bound on late-stage performance? In the case studies presented in the appendix, certain joints are fixed to 0, and specific coordinations are hardcoded as synergies. While such priors are highly beneficial for locomotion, might they conversely restrict performance in more complex manipulation tasks, perturbation recovery, or task switching?
3. LOAM-Race’s acquisition function is underspecified. Please specify the acquisition function in LOAM-Race precisely: what are the regression features, what is the prediction target, and what prior or regularization is used in BRR?
4. How does LOAM-Race compare empirically to widely used budgeted selection methods such as Successive Halving (SHA), Hyperband, Bayesian Optimization and Hyperband (BOHB), or Freeze-Thaw Bayesian Optimization?
5. LOAM-Race appears to avoid the difficulties caused by changing observation/action specifications mid-training, since each candidate is trained as its own policy-environment pair rather than reusing a single policy across interfaces. However, the paper does not analyze whether intermittent pause-resume training across candidates introduces additional optimization variance, nor does it clarify how optimizer state and any training state are handled across racing rounds.

**Limitations:**

yes

**Strengths And Weaknesses:**

**Strengths**

1. This paper focuses on the design of observation/action models, distinguishing itself from the prior work where LLMs are primarily utilized for reward design.
2. The experiments demonstrate that an appropriate redesign of the observation and action spaces can indeed yield superior performance.
3. The evaluation encompasses two benchmark suites (HumanoidBench and Isaac Lab), covering both locomotion and manipulation tasks, and provides comprehensive ablation studies.

**Weaknesses**

1. The method appears to rely strongly on structured semantic context, including task descriptions, variable names, shapes, and index mappings, and the paper does not evaluate anonymized or weakly semantic settings. This limits the current evidence for broad applicability.
2. Appendix F.4 suggests non-trivial sensitivity to prompt specification. Since performance improves after adding more expert guidance and clarifying implicit whole-body constraints, the framework may still depend materially on human-authored semantic scaffolding, which tempers its claim as a broadly automated design framework.
3. The action modeling component (e.g., low-dimensional action expansion, hardcoded synergies, and fixed-joint priors) appears central, but it is quantitatively under-specified. It remains unclear how latent action dimensions are chosen across tasks, or how much these action priors contribute, especially on manipulation tasks.

---

> ### Author Rebuttal · Authors · 2026-03-31
>
> We thank for the detailed review. We have conducted additional experiments available at https://sites.google.com/view/icml-loam.
>
> ### **[W1,Q1] Weakly semantic settings**
>
> LOAM does rely on semantically meaningful environment information, which we believe is essential for constructing meaningful observation and action spaces. Without semantic information (e.g., variable meanings, shapes, or task descriptions), constructing a meaningful observation/action interface becomes ill-posed, as neither LLMs nor human experts can interpret the variables. Therefore, our claim is not that LOAM operates without usable semantics. Rather, it is intended for environments where variables have meaningful descriptions. Under this assumption—where the same level of information available to human designers is provided—LOAM can automate the observation/action design process that would otherwise require manual engineering.
>
> We will revise the paper to make this boundary explicit: LOAM assumes semantically meaningful environment descriptions, and within this setting it automates observation/action design without manual feature engineering.
>
> ### **[W2] Prompt sensitivity**
>
> We thank the reviewer for this insightful observation. We agree that performance improves with clearer task descriptions, as shown in Appendix F.4. However, we emphasize that this does not reflect additional expert knowledge or hidden prompt engineering, but rather the removal of ambiguity in task specification. In the original description (e.g., “reach a target with the left hand”), the requirement of whole-body coordination is implicit but not explicitly stated. This can lead the LLM to misinterpret the task as involving only local motion. By clarifying such implicit constraints (e.g., maintaining stable posture), we make the intended task specification explicit.
>
> Importantly, this is not specific to LLMs. Even human experts require clear and unambiguous task definitions to design effective observation/action spaces. In this sense, the observed sensitivity reflects the importance of precise task specification rather than reliance on human-authored scaffolding. We therefore view this as a natural and desirable property: LOAM scales with the clarity of the provided task description, allowing users to improve performance by reducing ambiguity.
>
> ### **[W3,Q2] Action modeling design and expressivity**
>
> LOAM's action modeling is not limited to simple masking. The LLM determines *both the dimensionality and structure of the action space* from the task description and variable semantics, without human expert knowledge.
>
> This does not inherently impose an upper bound on performance. In locomotion tasks, the LLM may disable unnecessary joints, which can improve efficiency. But in manipulation, the opposite can happen: in Isaac-Lift-Cube-Franka-v0, the generated action space expands from 8D to 12D by adding group-control and posture-related channels, which can be found in the link (5. Example). This directly contradicts the concern that LOAM only compresses actions or enforces a restrictive prior. More broadly, LOAM does not use a fixed template for latent action dimensions across tasks; it can reduce, preserve, or expand the original action space depending on the task. We will clarify this mechanism and provide more quantitative details in the paper.
>
> ### **[Q3] Acquisition function specification**
>
> Thank you for pointing this out and we will clarify this in the revised version.
>
> **Features (X):** training session index. We use returns from the most recent `window_size=5` sessions.
>
> **Target (y):** individual evaluation returns. BRR predicts the expected return at a future step within the remaining budget (up to 50 sessions). With fewer than 5 points, pessimistic padding using the lowest-mean session prevents overestimation.
>
> **Prior/regularization :** scikit-learn `BayesianRidge` with default hyperparameters.
>
> **Selection:** UCB = predicted mean + predicted standard deviation.
>
> ### **[Q4] Comparison with other selection methods**
>
> We added SHA on Basic Locomotion tasks (stand, walk, run) (5 seeds) in the link (1. Comparison). LOAM-Race (w/ BRR) outperformed current-best selection and SHA.
>
> Due to time constraints, we use SHA as a representative baseline. Hyperband is a collection of SHA instances, so SHA provides a reasonable approximation. BOHB and Freeze-Thaw BO require similarity or density modeling across candidates, which is not well-defined in our setting where candidates are programmatic interfaces generated by an LLM.
>
> ### **[Q5] Pause-resume and state consistency**
>
> Each candidate maintains its full training state, including policy and optimizer. When resumed, the exact state is restored, making training equivalent to uninterrupted execution. Thus, pause-resume introduces no additional optimization variance. The overhead is negligible compared to training time.

---

> > ### Author Rebuttal · Reviewer_jmEC · 2026-04-02
> >
> > Thank you for your detailed rebuttal, which has addressed several of my earlier questions. However, one core concern remains insufficiently resolved: whether a fixed action mapping, once specified before training, may become a bottleneck for late-stage performance. I appreciate that the Isaac-Lift-Cube example usefully clarifies that LOAM does not merely compress the action space, yet my concern is different. In your run case study, the action model fixes hip_yaw to zero, ties shoulder_roll to hip_roll with a fixed coefficient, and zeros out all hand actuators. These priors may be sensible for straight-line running, but the current evidence does not show whether such frozen structure could limit performance in broader locomotion or manipulation settings.

---

> > > ### Author Response · Authors · 2026-04-02
> > >
> > > ### **[Concern: Fixed Action Mapping as a Potential Bottleneck]**
> > >
> > > We thank the reviewer for this thoughtful concern. We agree that a fixed action mapping can, in principle, become a bottleneck when the imposed structure is too restrictive for the task. This concern can arise even when the task is fixed, and becomes even more pronounced when the task changes during training. Accordingly, we do not claim that a single fixed mapping is universally optimal beyond the given task specification.
> > >
> > > **Our claim is limited to the fixed-task setting considered in this paper.** In that setting, our goal is to construct an action interface aligned with the task, rather than a heuristic for early learning. We did not design the prompt or action model to optimize only early-stage learning speed. Empirically, our main results (Figures 6 and 7) show that the learned policies improve not only early learning speed but also final return, rather than being limited by late-stage bottlenecks.
> > >
> > > Importantly, LOAM generates **task-specific** action models, conditioned on both the task and robot, rather than a generic or universal models. For example, in the run task, hand actuators are suppressed as they are not relevant to the specified objective. In contrast, in a broader manipulation task such as h1hand-truck-v0 (unloading packages from a truck onto a platform), the LLM constructs a richer action structure using a 39-dimensional latent action space: lower body, torso, and arms are largely preserved for stability and manipulation, while each hand is represented through synergy channels (e.g., wrist motion, thumb opposition, and coordinated finger flexion) that enable effective grasping. This demonstrates that LOAM adapts the action interface to the task.
> > >
> > > At the same time, we emphasize that even with task-specific modeling, this does not eliminate the possibility of generating overly restrictive or suboptimal action mappings for the given task. To mitigate this, we introduce LOAM-Race, which evaluates multiple candidate mappings under a shared budget. Some candidates learn quickly but plateau, while others improve more slowly but achieve higher final performance. LOAM-Race reallocates resources based on both current performance and predicted future potential, reducing the risk of over-committing to mappings that are only strong in early stages. We observe such behavior in practice (see link 4. Visualization), where training initially favors fast candidates but later shifts toward those with higher long-term potential. While this does not adapt a selected mapping online, it helps avoid committing the full budget to a suboptimal fixed interface.
> > >
> > > ---------------
> > >
> > > Dear Reviewer jmEC,
> > >
> > > We have provided detailed responses to your comments and would greatly appreciate any further feedback if you have the opportunity to review them, as the rebuttal period is approaching its final day. If any part of our response remains unclear or insufficient, we would be more than happy to provide additional clarification.
> > >
> > > Thank you for your time and consideration.

---

### Official Review · Reviewer_jbXo · 2026-03-13

**Soundness:** 3
**Presentation:** 2
**Significance:** 3
**Originality:** 3
**Overall Recommendation:** 4
**Confidence:** 4

**Summary:**

The paper proposes LOAM (LLM-based design of Observation and Action Models), a framework designed to automate the synthesis of observation and action representations in reinforcement learning. The approach avoids traditional, manually engineered mappings and leverages large language models to generate executable Python functions that define these models. Additionally, the paper presents LOAM-Race, an adaptive selection mechanism that evaluates multiple candidates concurrently, allocating training resources to the most promising options via upper-confidence-bound (UCB).

**Compliance With Llm Reviewing Policy:**

Affirmed.

**Final Justification:**

The rebuttal adequately addresses my concerns. The distinction between RL-level sparse reward and observation/action design is well argued. I maintain my positive score.

**Key Questions For Authors:**

1. have the authors tested on real world tasks beyond simulation?
2. Can iterative refinement of model spaces be useful in achieving improved performance? Such iterative refinement based has been exlored in iterative reward generation but not on observation/action spaces.

**Limitations:**

Please see weaknesses above.

**Strengths And Weaknesses:**

Strengths:
1. The paper tackles a well-known bottleneck in applying RL to complex robotic systems, which is the design of observation and action spaces. The proposed approach of using LLMs to automate this process is novel, and presents a compelling new direction for environment design in RL.
2. The experiment results in simulation are thorough and clearly show that LOAM and LOAM-Race consistently and significantly outperform strong baselines like FastTD3.
3. The paper provides complete templates, prompts, and descriptions which would help with reproducing the experimental benchmarks.

Weaknesses:
1. Although the LLMs are used to generate the code, the observation and action model prompts have to be structurally defined for the task, which requires human effort. For example, in the HumanoidBench environment, the success depends on access to pre-defined structure of the tasks, which may be rare in complex domains like non-robotics scenarios.
2. The human-engineered models prompts' requirement raises concerns on task generalizability, especially in environments with sparse rewards.

---

> ### Author Rebuttal · Authors · 2026-03-31
>
> We thank  for the constructive feedback.
>
> ### **[W1] Prompt structure, human effort, and applicability beyond robotics**
>
> We thank the reviewer for raising this important point. Our main clarification is that the required human effort is minimal, and LOAM's prompt structure is not specific to robotics.
>
> Most prompt components are fully reusable across tasks and environments. In practice, only two parts require adaptation: **Available Variable Info** and **Task Description**.
>
> | Component | Dependency | Human Effort |
> | --- | --- | --- |
> | Role / Objective | Independent | None; reusable across tasks and environments |
> | Instruction | Independent | None; reusable across tasks and environments |
> | Reasoning & Planning | Independent | None; reusable across tasks and environments |
> | Output Guide | Independent | None; reusable across tasks and environments |
> | Available Variable Info | Simulator / Robot-dependent | Extracted from simulator APIs or XML; straightforward |
> | Task Description | Task-dependent | Written per task; no expert knowledge required |
>
> **Available Variable Info** is built by collecting information already exposed by the environment, such as simulator documentation, APIs, or XML specifications, and formatting it into the prompt. In our experiments, we collected the full set of available variables from the documentation (e.g. https://isaac-sim.github.io/IsaacLab/main/source/api/lab/isaaclab.assets.html#isaaclab.assets.RigidObjectData) rather than selecting variables in a task-specific manner, so the process is not biased toward particular tasks. For a new simulator, the same collection step is needed, but it is straightforward and does not require domain expertise.
>
> **Task Description** only specifies the task objective, initialization, and termination conditions. Writing it does not require expert knowledge; a clear and unambiguous description of the task is sufficient for LOAM to generate effective observation/action models.
>
> Importantly, these two components provide only the same level of information that a human expert would also need in order to design observation and action spaces. LOAM automates this design process from that minimal input.
>
> LOAM is also applicable beyond robotics. Its key requirement is simply that the environment provides variables with meaningful semantic descriptions. This assumption holds not only for robotics simulators, but also for many other well-documented control environments.
>
> ### **[W2] Sparse reward generalizability**
>
> We appreciate this concern. LOAM's prompts are independent of whether the reward is sparse or dense. The prompt only requires a task description (including success conditions) and available variable information, neither of which depends on the reward structure. Given these inputs, LOAM leverages the LLM's common knowledge to perform task-relevant observation/action modeling regardless of reward density. While RL training itself may be more challenging in sparse-reward settings, this is an issue at the RL algorithm level rather than a limitation of observation/action design, and is therefore orthogonal to the scope of our work.
>
> ### **[Q1] Real-world task evaluation**
>
> We have not yet had the opportunity to validate LOAM on a physical robot, but we believe the framework is well suited for real-world deployment. In practice, sensor and actuator information obtained through frameworks such as ROS can be formatted according to our template, making LOAM directly applicable. We consider real-world validation an exciting and important direction for future work.
>
> ### **[Q2] Iterative refinement of model spaces**
>
> Thank you for the insightful question. We explored iterative refinement in Appendix F.3 and found it to be sensitive to the quality of the initial generation. When the initial model is poorly structured, later refinements tend to inherit and amplify these issues, making recovery difficult. In contrast, generating multiple independent candidates and selecting among them with LOAM-Race is more robust because it is not constrained by a single initialization. For this reason, we adopt the multi-candidate strategy. Exploring more effective refinement methods remains an interesting direction for future work.

---

> > ### Author Rebuttal · Reviewer_jbXo · 2026-04-04
> >
> > I thank the authors for their detailed responses. My concerns have been sufficiently addressed, and I will maintain my positive score.

---

> > > ### Author Response · Authors · 2026-04-06
> > >
> > > We sincerely thank you for your constructive feedback and for your careful review of our responses. We appreciate your positive assessment.

---

### Official Review · Reviewer_5Niq · 2026-03-20

**Soundness:** 2
**Presentation:** 3
**Significance:** 3
**Originality:** 2
**Overall Recommendation:** 3
**Confidence:** 4

**Summary:**

The authors propose to automate state and action space design using LLMs, which capture rich semantic information, to design the state and action space. They evaluate the framework on challenging simulated humanoid and manipulation tasks, showing improvements over a competing LLM-based design framework.

**Compliance With Llm Reviewing Policy:**

Affirmed.

**Final Justification:**

I accidentally swapped this review with another paper. Since then, I messaged the ACs of both papers and updated the official review with the original. Though I feel bad that the authors did not have a chance to properly give their rebuttal, I will maintain my score.

**Key Questions For Authors:**

What accounts for the large gaps in performance between LOAM, LOAM-hand, and LOAM-race (left,center Figure 6b)?

What are the direct benefits of using LLM-based state/action spaces as opposed to other methods (e.g., representation learning)?

**Limitations:**

Yes.

**Strengths And Weaknesses:**

Strengths

LOAM avoids handcrafted design of the observation/action space. All versions of LOAM show improvements over the baselines (backbone RL and competing LLM-based feature/action design framework) The experiments include extensive analysis of LOAM and its components on a wide range of challenging locomotion and manipulation tasks.

Suggested improvements

The agent observation and action space are based on that of the raw environment. Since the authors consider only simulation environments, which present state-based observations (e.g., proprioceptive, object, contact states). The appropriate RL formulation is MDP, unless the raw observations are camera, tactile, etc. observations. The authors make the claim that "the design of observation and action spaces remains relatively underexplored", which seems to disregard important literature in RL that leverages representation learning methods (and/or LLMs) to derive state/action spaces [1-7].

The positioning of this work needs further clarity. The authors evaluate against one baseline that augments handcrafted states and rewards with features generated by GPT-5. They also evaluate against their backbone RL algorithm, FastTD3. Demonstrating advantages over existing approaches that leverage representation learning (of state/action space) would increase the significance of this proposed work. The results in Figure 6 could be presented in a table while the learning curves can focus on comparing one (best) version of LOAM with the two baselines.

[1] LASER: Learning a Latent Action Space for Efficient Reinforcement Learning, Allshire et al., 2021

[2] INFOrmation Prioritization through EmPOWERment in Visual Model-Based RL, Bharadhwaj et al., 2022

[3] Efficient Planning in a Compact Latent Action Space, Jiang et al, 2022

[4] For SALE: State-Action Representation Learning for Deep Reinforcement Learning, Fujimoto et al., 2023

[5] AdaWorld: Learning Adaptable World Models with Latent Actions, Gao et al., 2025

[6] Learning with Language-Guided State Abstractions, Peng et al., 2024

[7] Inventing Relational State and Action Abstractions for Effective and Efficient Bilevel Planning, Silver et al., 2022

---

### Decision · Program_Chairs · 2026-04-30

**Decision:**

Accept (regular)

**Comment:**

The paper addresses an important and underappreciated problem in RL – automating observation and action space design – and provides a practical, well‑evaluated solution. The empirical results are strong, and the LOAM‑Race mechanism is a useful contribution.

The authors should address the following in the final version:
(1) explicitly acknowledge the reliance on semantic structure as a limitation;
2) include the rebuttal experiments (action-only ablation, LESR on Isaac Lab, SHA comparison) in the main paper or appendix;
(3) discuss related work on representation learning for state/action spaces more thoroughly;
(4) increase the number of seeds to at least 5 for all experiments.